# Electrodiffusion dynamics in the cardiomyocyte dyad at nano-scale resolution using the Poisson-Nernst-Planck (PNP) equations

**Karoline Horgmo Jæger** [iD]*, **Aslak Tveito**

Department of Computational Physiology, Simula Research Laboratory, Oslo, Norway

* karolihj@simula.no

## Abstract

During each heartbeat, a voltage wave propagates through the cardiac muscle, triggering action potentials in approximately two billion cardiomyocytes. This electrical activity ensures the coordinated contraction of the heart, which is essential for its pumping function. A key event in this process is the opening of voltage-gated calcium channels in the cell membrane, allowing calcium ions to enter the cardiac dyad and triggering a large-scale release of calcium ions from the sarcoplasmic reticulum through ryanodine receptors. This process is fundamental to cardiac function because calcium subsequently binds to troponin, initiating the conformational changes necessary for myofilament contraction. The cardiac dyad is characterized by a very small volume with steep ionic concentration gradients, which is challenging for detailed mathematical modeling. Traditionally, the dyadic calcium concentration has been approximated using spatially averaged values or modeled with reaction-diffusion equations. However, at the nanometer (nm) and nanosecond (ns) scales, such approximations may be insufficient. At this resolution, the Poisson-Nernst-Planck (PNP) system provides a detailed continuous representation of the underlying electrodiffusion dynamics. Here, we present a nano-scale computational model, representing dyad dynamics using the PNP system. Potassium, sodium, and calcium channels are incorporated in the cell membrane, along with the sodium-calcium exchanger. We demonstrate the formation of the Debye layer in the resting state and highlight how both diffusive and electrical effects are required to maintain this equilibrium. Additionally, we show that cross-species ion interactions in the dyad are electrical, and that diffusion models fail to capture this effect. Finally, we illustrate how the dyad width and diffusion coefficient influence local ionic concentrations and the timing of calcium arrival at the ryanodine receptors. These results provide new insights into the electrodiffusive properties of the dyad and clarify when solving the full PNP system is necessary for accurate modeling.

**Data availability statement:** All relevant data are contained in the paper and its Supporting information files. The simulation code is

publically available at Zenodo:
https://doi.org/10.5281/zenodo.15470900.

**Funding:** KHJ and AT were supported by the
Research Council of Norway funded project
SIMBER #322312 and the SUURPh program
funded by the Norwegian Ministry of Education
and Research. The funders had no role in study
design, data collection and analysis, decision to
publish, or preparation of the manuscript.

**Competing interests:** The authors have
declared that no competing interests exist.

## Author summary

Computational models of the heart typically focus on a single spatial scale. The bidomain model describes electrical activity at the millimeter scale, suitable for large-scale cardiac electrophysiology. More recently, cell-based models have gained attention, addressing clusters of cardiomyocytes at the micrometer scale. Here, we investigate an even smaller but crucial subcellular structure: the cardiac dyad, a region approximately 40 million times smaller than the entire cardiomyocyte. During each heartbeat, voltage-gated calcium channels open, leading to a sharp rise in calcium concentration within the dyad. This triggers a larger calcium release from internal stores, a process fundamental to cardiac contraction. Because this occurs at the nanometer scale, a precise mathematical description requires solving the Poisson-Nernst-Planck (PNP) system, which captures electrodiffusion dynamics. However, solving these equations is challenging due to steep concentration gradients and rapid timescales. We employ a computational model that enables significantly longer time steps than traditional methods, making detailed simulations feasible. This allows us to examine how ion channels influence the formation of the Debye layer and determine how quickly calcium waves travel across the dyad to reach the ryanodine receptors on the sarcoplasmic reticulum.

## 1. Introduction

The functioning of the heart depends on the coordinated activity of approximately two billion cardiomyocytes, [1]. Each heartbeat begins with the initiation of an action potential in the sinoatrial node, which propagates through the myocardium as a wave, depolarizing cardiomyocytes in a well-coordinated manner. At rest, cardiomyocytes maintain a transmembrane potential of approximately –80 mV, sustained by steep ionic gradients across the cell membrane. For example, the intracellular $K^+$ concentration is about 25 times higher than in the extracellular space. Since $K^+$ channels remain open during the resting phase, diffusion alone would lead to a depletion of intracellular $K^+$. However, electrical forces counteract this effect, maintaining the intracellular $K^+$ concentration. When a depolarization wave reaches a cardiomyocyte, the transmembrane potential depolarizes, leading to the opening of voltage-gated $Na^+$ channels and a rapid influx of $Na^+$ ions. This further depolarizes the membrane, triggering a cascade of channel, exchanger, and transporter activations that collectively generate the full action potential.

The fundamental roles of electrical and diffusive forces across the cell membrane are well established and fully integrated into mathematical models of the action potential, [2–4]. However, in the intra- and extracellular domains, ion dynamics are often approximated using constant concentrations and voltages, see, e.g., [5], or reaction-diffusion equations, [6–8], which account for diffusion and buffering effects but neglect electrical forces. These simplifications have been very important in developing successful models of the propagating electrochemical wave in cardiac tissue, [9–11]. While such models commonly represent tissue-scale dynamics at the millimeter level, recent efforts have refined them to cellular scales, [12–16], and even subcellular scales down to the micrometer range, [17–21]. The integration of multiple spatial scales have also been studied, see, e.g., [22–24].

Although the Poisson-Nernst-Planck (PNP) equations provide the most accurate continuous description of electrodiffusion, they are rarely solved due to the extreme numerical resolution they require, see, e.g., [25,26]. This raises a critical question: is it sufficient to account

for electrical forces only at the cell membrane, or do these forces also influence ion dynamics in close proximity to the membrane? It is well known that near the membrane, the Debye layer forms, where ion concentrations deviate from electroneutrality and therefore cannot be accurately modeled using constant values or reaction-diffusion equations that neglect electrical effects. However, the significance of these deviations for intracellular ion dynamics remains unclear. Do these local charge imbalances merely represent a minor perturbation, or do they have tangible consequences away from the membrane? More specifically, do electrical forces within the Debye layer significantly influence the dynamics of the cardiac dyad outside the Debye layer?

The purpose of this paper is to present an implementation of the PNP equations in the cardiac dyad and to analyze the electrodiffusive dynamics in this highly confined and physiologically crucial space. We solve the equations using a finite difference approach similar to [26], but here, we demonstrate how ion channels and exchangers commonly used in action potential models can be adapted to function within the PNP framework. We use the model to show how perturbations from electroneutrality decay rapidly and how the resting state is established along with its associated Debye layer. Additionally, we investigate the effects of opening individual ion channels and exchangers. For example, we investigate how the sodium-calcium exchanger (NCX) influences the dyadic dynamics when a nearby $Ca^{2+}$ channel is open and when the $Ca^{2+}$ channel is closed. Finally, we examine how the width of the dyad and the ionic diffusion coefficients affect the arrival time across the dyad of the $Ca^{2+}$ wave originating from $Ca^{2+}$ channels in the cell membrane.

## 2. Methods

### 2.1. The Poisson-Nernst-Planck (PNP) system

We apply a nano-scale computational model of electrodiffusion in the cardiomyocyte dyad based on the PNP system of equations. The system reads

$$\nabla \cdot (\varepsilon_r \varepsilon_0 \nabla \phi) = -\rho, \tag{1}$$

$$\frac{\partial c_k}{\partial t} = \nabla \cdot D_k \nabla c_k + \nabla \cdot \left( \frac{D_k z_k e}{k_B T} c_k \nabla \phi \right), \tag{2}$$

for each of the considered ion species $k$. In most of our simulations, we consider the ion species $k = \{Na^+, K^+, Ca^{2+}, Cl^-\}$, but in some of the first simpler examples we consider only $k = \{K^+, Cl^-\}$.

In the PNP system (1) and (2), $\phi$ is the electrical potential (in mV) and $c_k$ is the concentration of ions of type $k$ (in mM). Furthermore, $\varepsilon_r$ is the relative permittivity of the medium (unitless), $\varepsilon_0$ is the vacuum permittivity (in fF/m), $z_k$ is the valence of ion species $k$ (unitless), $D_k$ is the diffusion coefficient of ion species $k$ (in $nm^2$/ms), $e$ is the elementary charge (in C), $k_B$ is the Boltzmann constant (in mJ/K), and $T$ is the temperature (in K). Moreover, $\rho$ is the charge density (in $C/m^3$) defined as

$$\rho = \rho_0 + F \sum_k z_k c_k, \tag{3}$$

where $\rho_0$ is the background charge density (in $C/m^3$) and $F$ is Faraday's constant (in C/mol). The default values used for the model parameters in our simulations are given in Table 1.

**Table 1. Parameter values used in the simulations.** Here, $\Omega_e$ refers to the extracellular domain, $\Omega_m$ refers to the membrane domain, and $\Omega_i$ refers to the intracellular domain (see Fig 1). Note that the intracellular diffusion coefficients for Na$^+$, K$^+$ and Cl$^-$ are set up such that the ratio between the intracellular and extracellular diffusion coefficients are the same as for Ca$^{2+}$. Moreover, in the cell membrane ($\Omega_m$), the diffusion coefficient is set to zero for all ions. Electrodiffusion through channels and exchangers in the membrane is handled using local fluxes as explained in Sect 2.4.

| Parameter | Description | Value | Ref. |
|---|---|---|---|
| $F$ | Faraday's constant | 96485.3365 C/mol | [27] |
| $\varepsilon_0$ | Vacuum permittivity | 8854 fF/m | [27] |
| $\varepsilon_1$ | Relative permittivity, $\varepsilon_r$, in $\Omega_e$ and $\Omega_i$ | 80 | [28] |
| $\varepsilon_m$ | Relative permittivity, $\varepsilon_r$, in $\Omega_m$ | 2 | [28] |
| $D_{e,\text{Na}^+}$ | Diffusion coefficient for Na$^+$ in $\Omega_e$ | $1.33 \cdot 10^6$ nm$^2$/ms | [29] |
| $D_{e,\text{K}^+}$ | Diffusion coefficient for K$^+$ in $\Omega_e$ | $1.96 \cdot 10^6$ nm$^2$/ms | [29] |
| $D_{e,\text{Ca}^{2+}}$ | Diffusion coefficient for Ca$^{2+}$ in $\Omega_e$ | $0.71 \cdot 10^6$ nm$^2$/ms | [29] |
| $D_{e,\text{Cl}^-}$ | Diffusion coefficient for Cl$^-$ in $\Omega_e$ | $2.03 \cdot 10^6$ nm$^2$/ms | [29] |
| $D_{i,\text{Na}^+}$ | Diffusion coefficient for Na$^+$ in $\Omega_i$ | $0.37 \cdot 10^6$ nm$^2$/ms | |
| $D_{i,\text{K}^+}$ | Diffusion coefficient for K$^+$ in $\Omega_i$ | $0.55 \cdot 10^6$ nm$^2$/ms | |
| $D_{i,\text{Ca}^{2+}}$ | Diffusion coefficient for Ca$^{2+}$ in $\Omega_i$ | $0.20 \cdot 10^6$ nm$^2$/ms | [30] |
| $D_{i,\text{Cl}^-}$ | Diffusion coefficient for Cl$^-$ in $\Omega_i$ | $0.57 \cdot 10^6$ nm$^2$/ms | |
| $z_{\text{Na}^+}$ | Valence of Na$^+$ | 1 | |
| $z_{\text{K}^+}$ | Valence of K$^+$ | 1 | |
| $z_{\text{Ca}^{2+}}$ | Valence of Ca$^{2+}$ | 2 | |
| $z_{\text{Cl}^-}$ | Valence of Cl$^-$ | $-1$ | |
| $e$ | Elementary charge | $1.60217662 \cdot 10^{-19}$ C | [27] |
| $k_B$ | Boltzmann constant | $1.380649 \cdot 10^{-20}$ mJ/K | [27] |
| $T$ | Temperature | 310 K | |

**2.1.1. Incorporating calcium binding buffers.** As a large portion of the intracellular Ca$^{2+}$ ions in the cardiomyocyte are bound to Ca$^{2+}$ binding buffers, we extend the PNP model (1)–(3) to also represent two stationary Ca$^{2+}$ binding proteins, one with high affinity and one with low affinity, based on [31]. The parameters characterizing these buffers are provided in Table 2.

The binding of an ion, $k$, to a buffer, $j$, is represented by the flux

$$J_{B_{k,j}} = k_{\text{on}}^{k,j} c_k \left( B_{\text{tot}}^{k,j} - b_{k,j} \right) - k_{\text{off}}^{k,j} b_{k,j}, \tag{4}$$

where $b_{k,j}$ is the concentration of the ion bound to the buffer, $B_{\text{tot}}^{k,j}$ is the total buffer concentration, and $B_{\text{tot}}^{k,j} - b_{k,j}$ is the concentration of buffer $j$ with no ion bound. Furthermore, $k_{\text{on}}^{k,j}$ and $k_{\text{off}}^{k,j}$ are the rate constants of the Ca$^{2+}$ buffer binding reaction. Incorporating this reaction into the system (1)–(3), we get

$$\nabla \cdot (\varepsilon_r \varepsilon_0 \nabla \phi) = -\rho, \tag{5}$$

$$\frac{\partial c_k}{\partial t} = \nabla \cdot D_k \nabla c_k + \nabla \cdot \left( \frac{D_k z_k e}{k_B T} c_k \nabla \phi \right) - \sum_{j \in B_k} J_{B_{k,j}}, \tag{6}$$

$$\frac{\partial b_{k,j}}{\partial t} = J_{B_{k,j}}, \tag{7}$$

$$\rho = \rho_0 + F \sum_k z_k \left( c_k + \sum_j b_{k,j} \right), \tag{8}$$

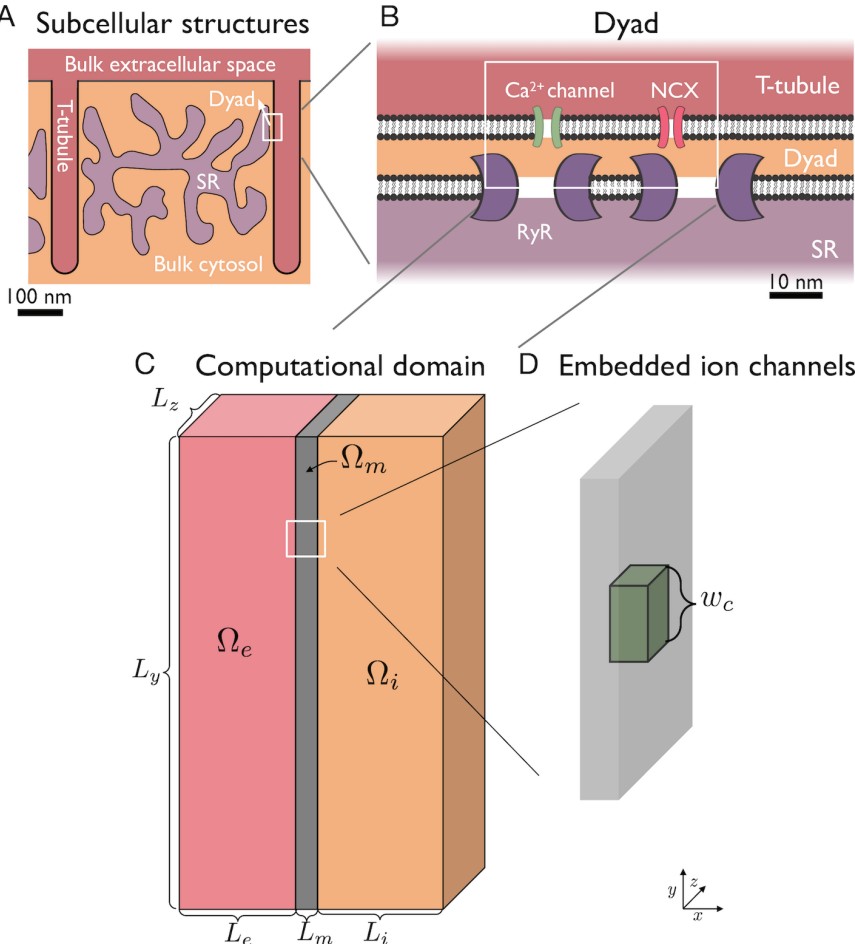

**Fig 1. Illustration of the considered domain.** A: In the cardiomyocyte, small domains called dyads are located in areas where the membrane of the SR is close to T-tubule membrane. B: In the dyad, Ca²⁺ channels and sodium-calcium exchangers (NCXs) in the cell membrane are directly apposed to ryadodine receptor channels (RyRs) in the membrane of the SR. C: The computational domain consists of an extracellular space, $\Omega_e$, a cell membrane, $\Omega_m$, and an intracellular domain, $\Omega_i$. A part of this domain is used to represent the dyad with an associated part of the cell membrane and extracellular space (T-tubule). D: In the computational domain, ion channels and exchangers occupy specific locations in the cell membrane.

**Table 2. Parameters characterizing the intracellular Ca²⁺ binding buffers.** The values are based on [31].

| Parameter | High affinity buffer | Low affinity buffer |
|---|---|---|
| $B_{\text{tot}}$ | 0.20 mM | 0.56 mM |
| $k_{\text{on}}$ | 100 ms⁻¹mM⁻¹ | 100 ms⁻¹mM⁻¹ |
| $k_{\text{off}}$ | 0.03 ms⁻¹ | 1.3 ms⁻¹ |

where $B_k$ is a collection of all the buffers to which ion $k$ may bind. In our computations, we consider two such Ca²⁺ binding buffers in the intracellular space ($\Omega_i$, see Fig 1) and no buffers for the remaining ions. Note here that in (8), $\rho_0$ is assumed to contain the charge of the buffer proteins. We apply the model (5)–(8) in all our PNP model simulations, except for some initial simple examples using (1)–(3). In the Supplementary Information (see S1 and S2 Figs), we compare solutions of the PNP model with and without Ca²⁺ binding buffers present.

## 2.2. Computational domain for dyad simulations

Except for the first simple examples, we consider a computational domain like the one illustrated in Fig 1C. This geometry is set up to represent:

1. The dyad (i.e., the intracellular cytosolic space between the cell membrane and the membrane of the sarcomplasmic reticulum (SR) in a location where these membranes are close and populated with $Ca^{2+}$ channels, sodium-calcium exchangers (NCXs) and ryanodine receptors (RyRs), see Fig 1A–B)
2. The cell membrane associated with the dyad (with embedded $Ca^{2+}$ channels and NCXs),
3. An associated part of the extracellular space (T-tubule).

The right boundary of the intracellular (dyad) part of the domain represents the SR membrane.

To represent the cell's resting state and a realistic upstroke duration, this geometrical setup is also extended to include part of the membrane (and the associated intracellular and extracellular spaces) that do not represent the dyad location, but is rather characterized as a part of the main cell membrane. In this main cell membrane part, we include a $K^+$ channel and a $Na^+$ channel.

The total computational domain is shaped as a rectangular cuboid, like illustrated in Fig 1C. The applied domain sizes are reported in Table 3. We let the intracellular part of the domain be denoted by $\Omega_i$, the extracellular part be denoted by $\Omega_e$, and the membrane part be denoted by $\Omega_m$.

As also observed in [26], the duration of the transmembrane dynamics (e.g., the duration of the upstroke) depends on the number of ion channels and the total membrane area included in the simulation. In our simulations, we consider one $K^+$ channel, one $Na^+$, one $Ca^{2+}$ channel and one NCX. For this setup, we found that a membrane area of $L_y \times L_z = 1000$ nm$\times$1000 nm was suitable to achieve a physiologically realistic upstroke time of about 0.5 ms.

**2.2.1. Boundary conditions.** For the electrical potential, $\phi$, we apply homogeneous Neumann boundary conditions on all boundaries, except for the leftmost boundary (see Fig 1C). On that boundary we use homogeneous Dirichlet boundary conditions ($\phi = 0$). For the ionic concentrations, we use homogeneous Neumann boundary conditions on all boundaries. These take the form

$$\left( D_k \nabla c_k + \left( \frac{D_k z_k e}{k_B T} c_k \nabla \phi \right) \right) \cdot n = 0, \tag{9}$$

**Table 3. Default geometry parameter values used in the simulations.** For definitions of the lengths, see Fig 1C. Note that $L_i$ denotes the length from the membrane of the T-tubule to the membrane of the SR (dyad width), and will be used as a control parameter in computational experiments below.

| Parameter | Value |
|---|---|
| $L_e$ | 100 nm |
| $L_m$ | 5 nm |
| $L_i$ | 7 nm |
| $L_y$ | 1000 nm |
| $L_z$ | 1000 nm |
| $w_c$ | 4 nm |

for all ion species, $k$, where $n$ is the outward pointing normal vector at the boundary. These boundary conditions amount to no flow of ions across the domain boundary. At the right-most boundary this represents that the membrane of the SR is closed for flow of ions. The remaining boundaries are located quite far from the ion channels and the considered dynamics and Neumann boundary conditions are chosen for simplicity and because they, unlike Dirichlet boundary conditions, do not interfere with the formation of the Debye layer when applied close to the membrane.

## 2.3. Initial conditions and background charge density

The initial conditions used for the ionic concentrations in the intracellular and extracellular spaces are provided in Table 4. The initial conditions for $Ca^{2+}$ bound to the two types of intracellular $Ca^{2+}$ binding proteins are set up such that the right-hand side of (4) is initially zero. The background charge density, $\rho_0$, is set up such that $\rho$ defined in (8) is zero for these initial conditions, hence the initial condition is always electroneutral. The background charge density is required to obtain physiological electroneutrality away from the membrane.

## 2.4. Representation of ion channels and exchangers

One approach for representing ion channels embedded in the cell membrane is by selectively allowing for electrodiffusion in the part of the membrane that represents the channel (see Fig 2A). For instance, for a $K^+$ channel, the diffusion coefficient is zero for all ions except for $K^+$ and set to some scaled parameter $d_{K^+}$ (in $nm^2/ms$) for $K^+$ ions. This approach for representing ion channels was applied in [26]. One disadvantage with this approach is that it might not be evident how to determine suitable diffusion coefficients or background charge densities ($\rho_0$) for the ion channels (see [26]). Another disadvantage is that this approach is only straightforwardly implemented for ion channels, and not for membrane pumps and exchangers, like, e.g., the NCX.

To represent the fluxes through pumps and exchangers, it is more convenient to represent the fluxes as internal boundary conditions between the membrane and the intracellular and extracellular spaces (see Fig 2B). In this case, a specified expression may be directly used for the flux instead of letting the flux through the channel be governed by the PNP equations. To represent channels and exchangers in a similar manner, we will apply this alternative approach to both ion channel and exchanger fluxes in this study. However, in the Supplementary Information, we observe that the two alternative approaches illustrated in Fig 2 provide quite similar solutions (see S1 Appendix).

**Table 4. Initial conditions for the ionic concentrations in the intracellular and extracellular domains.** The values are based on [29,32,33].

| Ion | Extracellular | Intracellular |
|---|---|---|
| $Na^+$ | 100 mM | 12 mM |
| $K^+$ | 5 mM | 125 mM |
| $Ca^{2+}$ | 1.4 mM | 0.0001 mM |
| $Cl^-$ | 107.8 mM | 15 mM |

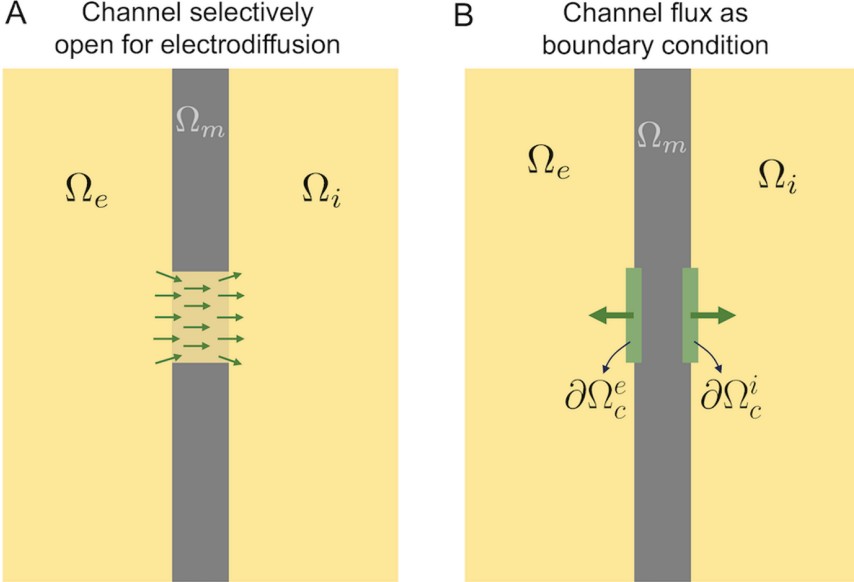

**Fig 2. Illustration of two alternative representation of ion channels embedded in the membrane.** A: Ion channels are represented as subdomains of the membrane which selectively allow for electrodiffusion of certain ion species. For example, a $K^+$ channel only allows for electrodiffusion of $K^+$ ions. B: The flux through ion channels are represented as internal boundary conditions for the ion concentrations associated with the channel. In this representation, electrodiffusion in the channel is formulated as a boundary condition with a specified flux. This flux can be defined by the integration of the 1D Nernst equation for the specific ion specie under consideration.

The internal boundary conditions illustrated in Fig 2B take the form

$$\left( D_k \nabla c_k + \frac{D_k z_k e}{k_B T} c_k \nabla \phi \right) \cdot n_i = -J, \text{ at } \partial\Omega_c^i, \tag{10}$$

$$\left( D_k \nabla c_k + \frac{D_k z_k e}{k_B T} c_k \nabla \phi \right) \cdot n_e = J, \text{ at } \partial\Omega_c^e, \tag{11}$$

where $\partial\Omega_c^i$ and $\partial\Omega_c^e$ are the interfaces between the membrane and the intracellular and extracellular domains, respectively, at the location of the channel (see Fig 2B), and $n_i$ and $n_e$ are the outward pointing unit normal vectors of the intracellular and extracellular domains, respectively. Furthermore $J$ is an expression for the channel or exchanger flux (in mMnm/ms). This flux is defined to be positive for a flux of ions in the direction from the intracellular to the extracellular space.

**2.4.1. Ion channel flux.** The current through a single open ion channel (in yA = $10^{-24}$A) is often represented using the model

$$i_k = g_k(v - v_{0,k}), \tag{12}$$

where $g_k$ is the conductance of a single open channel (in zS = $10^{-21}$S), $v$ is the transmembrane potential (in mV), and $v_{0,k}$ is the Nernst equilibrium potential of the channel (in mV), see, e.g., [34]. This equilibrium potential is given by

$$v_{0,k} = \frac{k_B T}{z_k e} \ln\left( \frac{c_{k,e}}{c_{k,i}} \right), \tag{13}$$

where $c_{k,e}$ and $c_{k,i}$ are the concentrations of ion species $k$ at the extracellular and intracellular sides of the channel, respectively. This single channel current may be converted to a single channel flux (in ymol/s) by dividing the current by the ion species valence, $z_k$, and Faraday's constant, $F$:

$$j_k = \frac{i_k}{z_k F} = \frac{1}{z_k F} g_k (v - v_{0,k}). \tag{14}$$

Furthermore, the single channel flux may be converted to a single channel flux density (in ymol/($\text{nm}^2$s) = mMnm/ms) by dividing the flux by the channel area (given by $w_c^2$, in $\text{nm}^2$, see Fig 1B):

$$J_k = \frac{j_k}{w_c^2} = \frac{1}{z_k F w_c^2} g_k (v - v_{0,k}). \tag{15}$$

The single channel conductances, $g_k$, used for the ion channels in our simulations are provided in Table 5. The transmembrane potential, $v$, is computed by

$$v = \phi_i - \phi_e, \tag{16}$$

where $\phi_i$ is the average electrical potential on $\partial\Omega_c^i$, and $\phi_e$ is the average electrical potential on $\partial\Omega_c^e$ (the inlet and outlet of the channels, see Fig 2B). The concentrations $c_{k,e}$ and $c_{k,i}$ in the Nernst equilibrium potential (13) are similarly defined as the average ionic concentrations at $\partial\Omega_c^e$ and $\partial\Omega_c^i$, respectively.

**2.4.2. The sodium calcium exchanger flux.** For the NCX flux, we use the formulation of the current density, $I_{\text{NCX}}$, from [37], i.e.,

$$I_{\text{NCX}} = \bar{I}_{\text{NCX}} \frac{\exp\left(\frac{vev}{k_B T}\right)(c_{\text{Na}^+,i})^3 c_{\text{Ca}^{2+},e} - \exp\left(\frac{(v-1)ev}{k_B T}\right)(c_{\text{Na}^+,e})^3 c_{\text{Ca}^{2+},i}}{s_{\text{NCX}} \left(1 + \left(\frac{K_{\text{act}}}{c_{\text{Ca}^{2+},i}}\right)^2\right)\left(1 + k_{\text{sat}} \exp\left(\frac{(v-1)ev}{k_B T}\right)\right)}, \tag{17}$$

$$s_{\text{NCX}} = K_{\text{Ca},i}(c_{\text{Na}^+,e})^3 \left(1 + \left(\frac{c_{\text{Na}^+,i}}{K_{\text{Na},i}}\right)^3\right) + K_{\text{Na},e}^3 c_{\text{Ca}^{2+},i}\left(1 + \frac{c_{\text{Ca}^{2+},i}}{K_{\text{Ca},i}}\right)$$
$$+ K_{\text{Ca},e}(c_{\text{Na}^+,i})^3 + (c_{\text{Na}^+,i})^3 c_{\text{Ca}^{2+},e} + (c_{\text{Na}^+,e})^3 c_{\text{Ca}^{2+},i}. \tag{18}$$

The ionic concentrations $c_{\text{Na}^+,i}$, $c_{\text{Na}^+,e}$, $c_{\text{Ca}^{2+},i}$, and $c_{\text{Ca}^{2+},e}$ are defined as the average concentrations at $\partial\Omega_c^i$ and $\partial\Omega_c^e$ like for the ion channel fluxes. The transmembrane potential, $v$, is also defined by (16) in the same manner as for the ion channel fluxes. The parameter values involved in the channel flux are given in Table 5.

**Table 5. Parameters for the channel and exchanger fluxes.** The single channel conductances, $g_{\text{K}^+}$, $g_{\text{Na}^+}$, and $g_{\text{Ca}^{2+}}$ are taken from [35]. The NCX density, $\delta_{\text{NCX}}$, is taken from [36] and the remaining parameters characterize the NCX and are taken from [37].

| Parameter | Value | Parameter | Value |
|---|---|---|---|
| $g_{\text{K}^+}$ | 5 pS = $5 \cdot 10^9$ zS | $k_{\text{sat}}$ | 0.3 |
| $g_{\text{Na}^+}$ | 20 pS = $20 \cdot 10^9$ zS | $K_{\text{act}}$ | 0.00015 mM |
| $g_{\text{Ca}^{2+}}$ | 8 pS = $8 \cdot 10^9$ zS | $K_{\text{Ca},i}$ | 0.0036 mM |
| $\bar{I}_{\text{NCX}}$ | 4.9 A/F | $K_{\text{Ca},e}$ | 1.3 mM |
| $\delta_{\text{NCX}}$ | $4 \cdot 10^{-4}$ $\text{nm}^{-2}$ | $K_{\text{Na},i}$ | 12.3 mM |
| $C_m$ | 10 000 yF/$\text{nm}^2$ | $K_{\text{Na},e}$ | 87.5 mM |
| $v$ | 0.3 | | |

The current density, $I_{\text{NCX}}$, represents the average current density over the membrane capacitance of a cardiomyocyte in units of A/F. The total current for the cell (in yA) is

$$I_{\text{NCX}}^{\text{tot}} = C_m A_m I_{\text{NCX}}, \tag{19}$$

where $C_m$ is the specific membrane capacitance (in yF/nm$^2$) and $A_m$ is the total membrane area (in nm$^2$). Assuming that the membrane contains $m$ NCXs, the current (in yA) through a single NCX is given by

$$i_{\text{NCX}} = \frac{I_{\text{NCX}}^{\text{tot}}}{m} = \frac{A_m C_m}{m} I_{\text{NCX}} = \frac{C_m}{\delta_{\text{NCX}}} I_{\text{NCX}}, \tag{20}$$

where

$$\delta_{\text{NCX}} = \frac{m}{A_m}, \tag{21}$$

is the density of NCXs on the cell membrane.

Like for the ion channels, we may now convert this single NCX current to single NCX fluxes (in ymol/s). The current $i_{\text{NCX}}$ is defined to be positive for a net positive flow of ions out of the cell. The NCX exchanges three Na$^+$ ions for one Ca$^{2+}$ ion. This means that when $i_{\text{NCX}}$ is positive, Na$^+$ moves out of the cell and Ca$^{2+}$ moves into the cell, and we get

$$j_{\text{Na}^+} = \frac{3}{F} i_{\text{NCX}}, \tag{22}$$

$$j_{\text{Ca}^{2+}} = -\frac{1}{F} i_{\text{NCX}}. \tag{23}$$

By dividing by the channel area, $w_c^2$, we get the flux densities (in ymol/(nm$^2$s) = mMnm/ms):

$$J_{\text{Na}^+} = \frac{j_{\text{Na}^+}}{w_c^2} = \frac{3}{F w_c^2} i_{\text{NCX}} = \frac{3 C_m}{F w_c^2 \delta_{\text{NCX}}} I_{\text{NCX}}, \tag{24}$$

$$J_{\text{Ca}^{2+}} = \frac{j_{\text{Ca}^{2+}}}{w_c^2} = -\frac{1}{F w_c^2} i_{\text{NCX}} = -\frac{C_m}{F w_c^2 \delta_{\text{NCX}}} I_{\text{NCX}}, \tag{25}$$

where $I_{\text{NCX}}$ is defined in (17) and the parameters are found in Table 5.

## 2.5. Numerical methods

We solve the system (5)–(8) using a finite difference discretization similar to what was done in [26]. However, instead of solving (5) and (6) in two different steps, we apply a coupled scheme in this study. This allows us to increase the time step from $\Delta t = 0.02$ ns used in [26] to $\Delta t = 1000$ ns. Convergence of the numerical scheme is shown in the Supplementary Information (see S2 Appendix).

**2.5.1. Temporal operator splitting.** We solve the system (5)–(8) using a first order temporal operator splitting technique (see, e.g., [38]) to split the buffering dynamics from the remaining system of equations. That is, for each time step, we first update $c_k$ and $b_k$ by solving

the non-linear ordinary differential equation (ODE) system

$$\frac{\partial c_k}{\partial t} = -\sum_{j \in B_k} J_{B_{k,j}}, \tag{26}$$

$$\frac{\partial b_{k,j}}{\partial t} = J_{B_{k,j}}, \tag{27}$$

and, next, in a second step, we update $c_k$ and compute $\phi$ by solving the partial differential equation (PDE) system

$$\nabla \cdot (\varepsilon_r \varepsilon_0 \nabla \phi) = -\left( \rho_0 + F \sum_k z_k \left( c_k + \sum_j b_{k,j} \right) \right), \tag{28}$$

$$\frac{\partial c_k}{\partial t} = \nabla \cdot D_k \nabla c_k + \nabla \cdot \left( \frac{D_k z_k e}{k_B T} c_k \nabla \phi \right). \tag{29}$$

More specifically, at each time step, $n$, we assume that the solutions from the previous time step, $c_k^{n-1}$ and $b_{k,j}^{n-1}$, are known and use an explicit forward Euler discretization of (26) and (27) to compute $\tilde{c}_k^n$ and $b_{k,j}^n$:

$$\frac{\tilde{c}_k^n - c_k^{n-1}}{\Delta t} = -\sum_{j \in B_k} J_{B_{k,j}}^{n-1}, \tag{30}$$

$$\frac{b_{k,j}^n - b_{k,j}^{n-1}}{\Delta t} = J_{B_{k,j}}^{n-1}. \tag{31}$$

Here, $\tilde{c}_k^n$ are interim solutions to be applied in the next step of the operator splitting scheme, and $b_{k,j}^n$ are the updated solutions of $b_{k,j}$ for the current time step, $n$.

In the next step, we solve the system (28) and (29), in a coupled manner using an implicit backward Euler temporal discretization. To linearize the resulting system of equations, we approximate $c_k$ in the term $\nabla \cdot \left( \frac{D_k z_k e}{k_B T} c_k \nabla \phi \right)$ in (29) using the solution from the previous operator splitting step. This results in the following scheme for computing $\phi^n$ and $c_k^n$:

$$\nabla_h \cdot (\varepsilon_r \varepsilon_0 \nabla_h \phi^n) = -\left( \rho_0 + F \sum_k z_k \left( c_k^n + \sum_j b_{k,j}^n \right) \right), \tag{32}$$

$$\frac{c_k^n - \tilde{c}_k^n}{\Delta t} = \nabla_h \cdot D_k \nabla_h c_k^n + \nabla_h \cdot \left( \frac{D_k z_k e}{k_B T} \tilde{c}_k^n \nabla_h \phi^n \right). \tag{33}$$

Here, $\nabla_h$ is a finite difference discretization of $\nabla$ (see, e.g., [26]). Furthermore, $\tilde{c}_k^n$ is the solution from the first step of the operator splitting scheme. If no buffers for ion $k$ are present, $\tilde{c}_k^n$ is replaced by $c_k^{n-1}$.

**2.5.2. Discretization of membrane fluxes.** For the ion channel fluxes (15), we incorporate an implicit representation of the ionic concentrations, $c_k$, and the electrical potential, $\phi$, in the second step of the operator splitting scheme. To get a linear system, we approximate the logarithm term in the Nernst equilibrium potential (13) by a Taylor series approximation around the solutions from the first operator splitting step:

$$\ln(c_k^n) \approx \ln(\tilde{c}_k^n) + \frac{1}{\tilde{c}_k^n} (c_k^n - \tilde{c}_k^n) = \ln(\tilde{c}_k^n) - 1 + \frac{c_k^n}{\tilde{c}_k^n}, \tag{34}$$

where $c_k^n$ denotes the concentration of ion species $k$ at time step $n$, and $\tilde{c}_k^n$ is the solution from the first operator splitting step. This yields the approximation

$$\ln\left(\frac{c_{k,e}^n}{c_{k,i}^n}\right) = \ln(c_{k,e}^n) - \ln(c_{k,i}^n) \tag{35}$$

$$\approx \ln(\tilde{c}_{k,e}^n) - 1 + \frac{c_{k,e}^n}{\tilde{c}_{k,e}^n} - \ln(\tilde{c}_{k,i}^n) + 1 - \frac{c_{k,i}^n}{\tilde{c}_{k,i}^n} \tag{36}$$

$$= \ln\left(\frac{\tilde{c}_{k,e}^n}{\tilde{c}_{k,i}^n}\right) + \frac{c_{k,i}^n}{\tilde{c}_{k,e}^n} - \frac{c_{k,i}^n}{\tilde{c}_{k,i}^n}. \tag{37}$$

The flux (15) is thus expressed as

$$J_k = \frac{1}{z_k F w_c^2} g_k \left( \phi_i^n - \phi_e^n - \frac{k_B T}{z_k e} \left( \ln\left(\frac{\tilde{c}_{k,e}^n}{\tilde{c}_{k,i}^n}\right) + \frac{c_{k,i}^n}{\tilde{c}_{k,e}^n} - \frac{c_{k,i}^n}{\tilde{c}_{k,i}^n} \right) \right). \tag{38}$$

For the NCX fluxes (24) and (25), we use an explicit representation of the potential and ionic concentrations. In other words, we apply the concentrations from the previous operator splitting step, $\tilde{c}_k^n$, and the electrical potential from the previous time step, $\phi^{n-1}$.

**2.5.3. Mesh.**   To reduce the computational costs, we use a simple adaptive meshing approach, like in [26]. In this approach, a high resolution is applied near the membrane and the ion channels and a coarser resolution is applied elsewhere. A slice of the mesh in the $x$- and $y$- directions is illustrated in Fig 3. Near the membrane and the right boundary of the domain (representing the SR membrane), the distance between grid points is 0.5 nm in the $x$-direction, and the distance doubles for each grid point further away from the membrane. Similarly, the distance between grid points is 2 nm in the $y$- and $z$-directions near the channels or exchangers and doubles for each grid point as the distance from the channels or exchangers increases.

## 3. Results

In this section, we present the results of simulations using the model described above. The central goal is to investigate the electrodiffusive dynamics in the dyadic space of cardiomyocytes and to examine whether electrical effects, in addition to diffusion, significantly influence $Ca^{2+}$ transport across the dyad.

The key quantity of interest is the time required for $Ca^{2+}$ ions, entering through membrane channels, to reach the opposing RyRs on the SR membrane. However, before addressing this, we start with two simplified examples that illustrate general properties of the PNP system, such as the rapid decay towards electroneutrality and the formation of the Debye layer. These initial examples provide insights of the dynamics that also govern the dyadic space.

Next, we examine dyad-specific simulations, including scenarios where different ion channels and exchangers are opened. We analyze how the diffusion coefficient and dyad width affect local ionic concentrations and the $Ca^{2+}$ arrival time at the SR membrane. Finally, we compare the PNP model predictions with those of alternative modeling approaches, including reaction-diffusion equations and ODE-based compartment models, to clarify when the full PNP system is required.

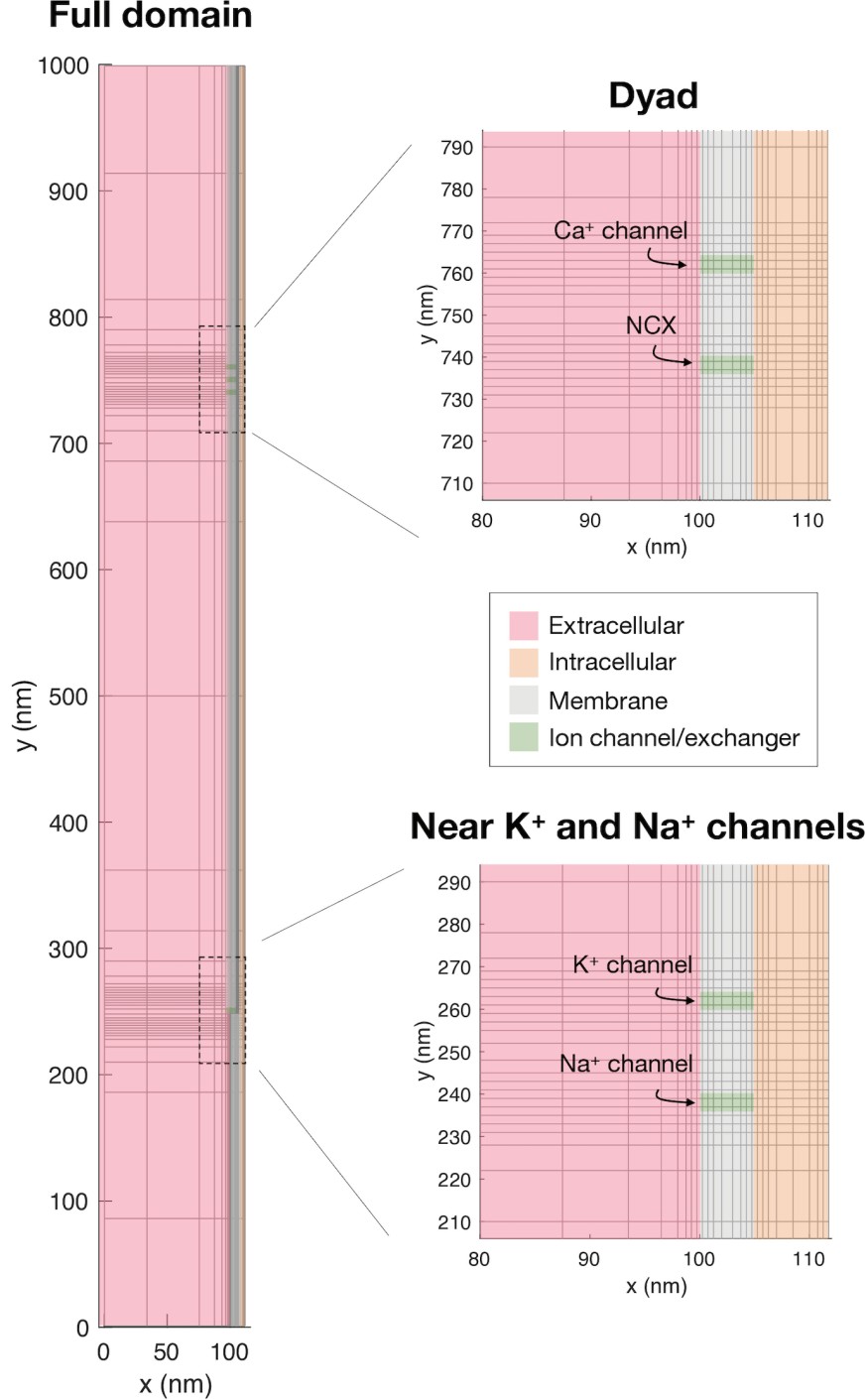

**Fig 3. Illustration of the mesh applied in the simulations.** We show a slice of the mesh in the $x$- and $y$-directions. The mesh is refined near the membrane and near membrane channels and exchangers. Similar refinements are applied in the $z$-direction. The $Ca^{2+}$ channel and the $Na^+/Ca^{2+}$-exchanger (NCX) are located in a volume referred to as the dyad. A $K^+$ and a $Na^+$ channel are located in another volume of interest in the simulations.

## 3.1. PNP model solutions of decay following perturbations from electroneutrality

To illustrate properties of the PNP model, we first consider a simple example of a perturbation from electroneutrality. We consider the PNP model (1)–(3) in two dimensions (2D) for the two ions $K^+$ and $Cl^-$. We let $\rho_0 = 0$ in the entire domain. At the boundary of the domain, we apply Dirichlet boundary conditions fixing the potential, $\phi$, at 0 mV and both ionic concentrations at 100 mM. The initial conditions are illustrated in Fig 4A. For these initial conditions, the concentrations do not initially fulfill electroneutrality (i.e., we have $\rho \neq 0$, where $\rho$ is defined in (3)).

Fig 4B shows the concentration solutions in the point in the center of the domain for the two ionic species as functions of time. We observe that for the first few nanoseconds, the dynamics are fast and that the dynamics are slower in the later part of the simulation. Fig 4C similarly shows the charge density, $\rho$ (defined in (3)), in the center point as a function of time. Here, we observe that the solutions approach electroneutrality ($\rho = 0$) fast during the first 3 ns of the simulation.

In Fig 4D we show the magnitude of the two terms $B_d = \nabla \cdot D_k \nabla c_k$ and $B_e = \nabla \cdot \left( \frac{D_k z_k e}{k_B T} c_k \nabla \phi \right)$ in (2) in the center point for the first few nanoseconds (left panel) and later in the simulation (right panel). We observe that during the first few nanoseconds (before electroneutrality is achieved) the electrical term, $B_e$, is much more prominent than the diffusional term, $B_d$. This electrical term drives the system fast towards electroneutrality during the first few nanoseconds of the simulation. The concentration of $K^+$ decreases and the concentration of $Cl^-$ increases. Next, after electroneutrality is achieved, the electrical term, $B_e$, is negligible, and the diffusional term, $B_d$, is most prominent. This terms slowly brings the concentrations towards the constant value of 100 mM.

## 3.2. Formation of a Debye layer near a membrane in a simple PNP model example

A second simple 2D example illustrating properties of the PNP model is displayed in Fig 5. In this case, we introduce a membrane between two domains, $\Omega_L$ and $\Omega_R$ (see Fig 5C). In the membrane, the diffusion coefficient is set to zero. We again consider the PNP system (1)–(3) for two ions, $K^+$ and $Cl^-$. We use homogeneous Neumann boundary conditions on all boundaries for both the electrical potential and the concentrations, except that we use the Dirichlet boundary condition $\phi = 0$ mV for the potential on the left boundary. The initial concentration is set to 100 mM for $Cl^-$ in both $\Omega_L$ and $\Omega_R$, but for $K^+$, it is set to is 100.1 mM in $\Omega_L$ and 99.9 mM in $\Omega_R$. This entails that $\rho$ is initially slightly positive in $\Omega_L$ and slightly negative in $\Omega_R$. In Fig 5A, we show the solutions near the membrane as functions of $x$ for five different points in time. We observe that a layer of adjusted concentrations and non-zero $\rho$ gradually forms on each side of the membrane. This layer near the membrane is often referred to as the Debye layer. In Fig 5B, we plot $\rho$ as a function of time in one point near the membrane and one point 20 nm right of the membrane. These points are marked as $x_1$ and $x_2$ in Fig 5C. We observe that away from the membrane, the solutions approach electroneutrality (i.e., $\rho$ approaches zero). Near the membrane, on the other hand, the magnitude of $\rho$ increases with time for the first few nanoseconds and then remains constant with time after the Debye layer has been formed.

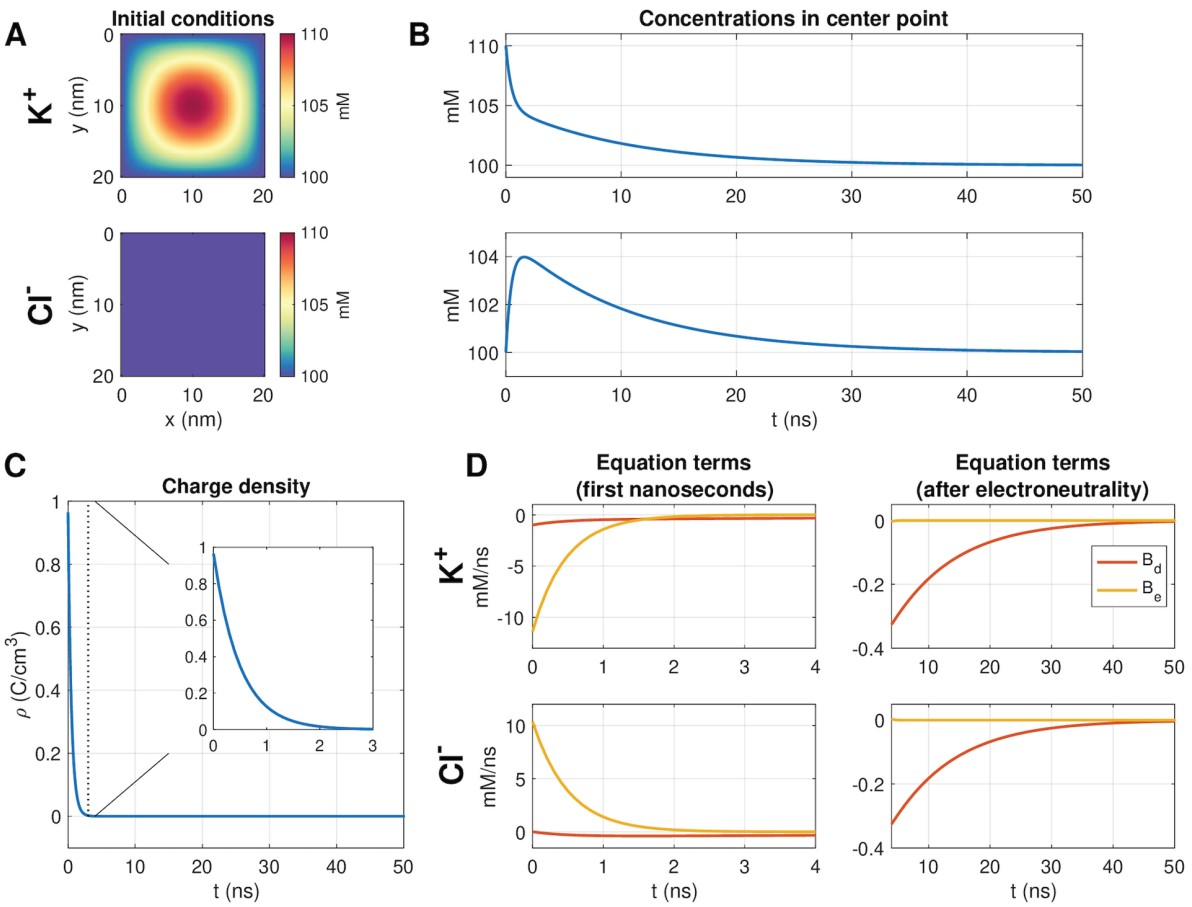

**Fig 4. Decay following a perturbation of the $K^+$ concentration in a PNP simulation.** We consider a simple 2D example with two ions, $K^+$ and $Cl^-$. A: Initial conditions for the two ions. B: Concentrations in the center point as functions of time. C: Charge density, $\rho$, in the center point as a function of time. D: The equation terms, $B_d = \nabla \cdot D_k \nabla c_k$ and $B_e = \nabla \cdot \left( \frac{D_k z_k e}{k_B T} c_k \nabla \phi \right)$, of (2) in the center point as functions of time during the first 4 ns of simulation (left) and after 4 ns (right). We use $D_{K^+} = D_{Cl^-} = 2.03 \cdot 10^6$ nm$^2$/ms, $\Delta t = 0.1$ ns and a uniform mesh with $\Delta x = \Delta y = 0.25$ nm.

## 3.3. Visualization of dyad simulation results

In this section, we describe the setup used to visualize the results of the dyad simulations in the following sections. To this end, Fig 6 illustrates the general setup used in the simulation visualizations.

**A: Titles.** In the upper (title) panel of the figure, each of the considered ionic species are listed, as well as $\rho$ and $\phi$. All figure panels below these titles depict the concentration of the ion species, or $\rho$ or $\phi$, named in these upper titles.

**B: Solution snapshots.** In the next two plot rows, snapshots of the solutions are displayed. The plot titles (Panel A) describes the variable displayed in each column. For each plot row, one point in time is considered. This time point is reported on the left side of each row. In the example illustration in Fig 6, only two time points are included ($t = 0$ ms and $t = 1$ ms), but for the actual result figures five time points are generally included.

The snapshots show the solutions in a sheet in the $x$-$y$-plane in the center of the domain in the $z$-direction. Moreover, we only consider a part of the domain that is in close proximity to channels of interest. Fig 6 shows the dynamics close to the $K^+$ and $Na^+$ channels,

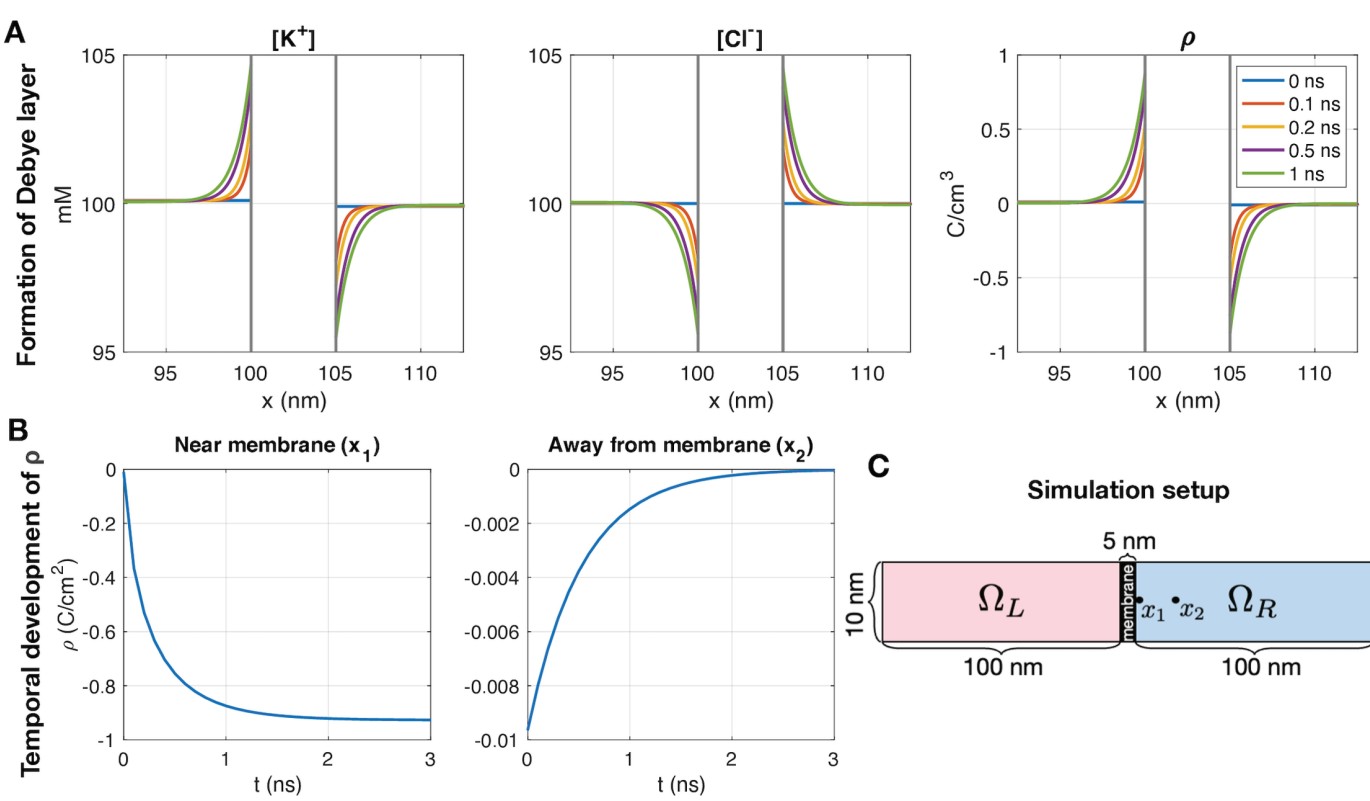

**Fig 5. Formation of a Debye layer near a membrane in a PNP simulation.** We consider a simple 2D example with two ions. Initially, the concentrations are constant in $\Omega_L$ and $\Omega_R$. The initial concentration of Cl$^-$ is 100 mM in both $\Omega_L$ and $\Omega_R$, and the initial concentration of K$^+$ is 100.1 mM in $\Omega_L$ and 99.9 mM in $\Omega_R$. A: Ionic concentrations and charge density near the membrane as functions of $x$ for five different points in time. B: Charge density, $\rho$, 0.1 nm right of the membrane ($x_1$) and 20 nm to the right of the membrane ($x_2$) as functions of time. C: Illustration of the simulation setup. We use $D_{K^+} = D_{Cl^-} = 2.03 \cdot 10^6$ nm$^2$/ms, $\Delta t = 0.1$ ns and a uniform mesh with $\Delta x = \Delta y = 0.1$ nm.

but some of the result figures focus on the dyad area (close to the Ca$^{2+}$ channel and NCX) instead.

In the plots of the ionic concentrations, the part of the domain representing the membrane is removed from the visualizations (because all concentrations are zero in the membrane). Instead, illustrations of the ion channels and exchangers are included. In these illustrations, a green color indicates that the channel is open and a red/pink color indicates that the channel is closed. Furthermore, for the ionic species that is able to move through the channel or exchanger, an arrow indicating the direction of flow of the ions is also included. The extracellular space is illustrated on the left side of the channels and the intracellular space is illustrated on the right side of the channels. For $\rho$ and $\phi$ the membrane is included in the solution snapshots. Thus, we do not explicitly display the ion channels and exchangers in these plots, but they are located at the locations indicated in the concentrations plots.

**C: Colorbars.** Below the solution snapshots, colorbars for these solutions are included. For $\phi$ and $\rho$ the same colormap scaling is used in the extracellular, intracellular, and membrane domains, and thus only one colorbar is included for each of these two variables. For the ionic concentrations, on the other hand, the colormap scaling is different in the intracellular and extracellular domains. Therefore, two colorbars are provided for each ionic species. The right colorbar shows the colormap scaling in the extracellular space and the left colorbar

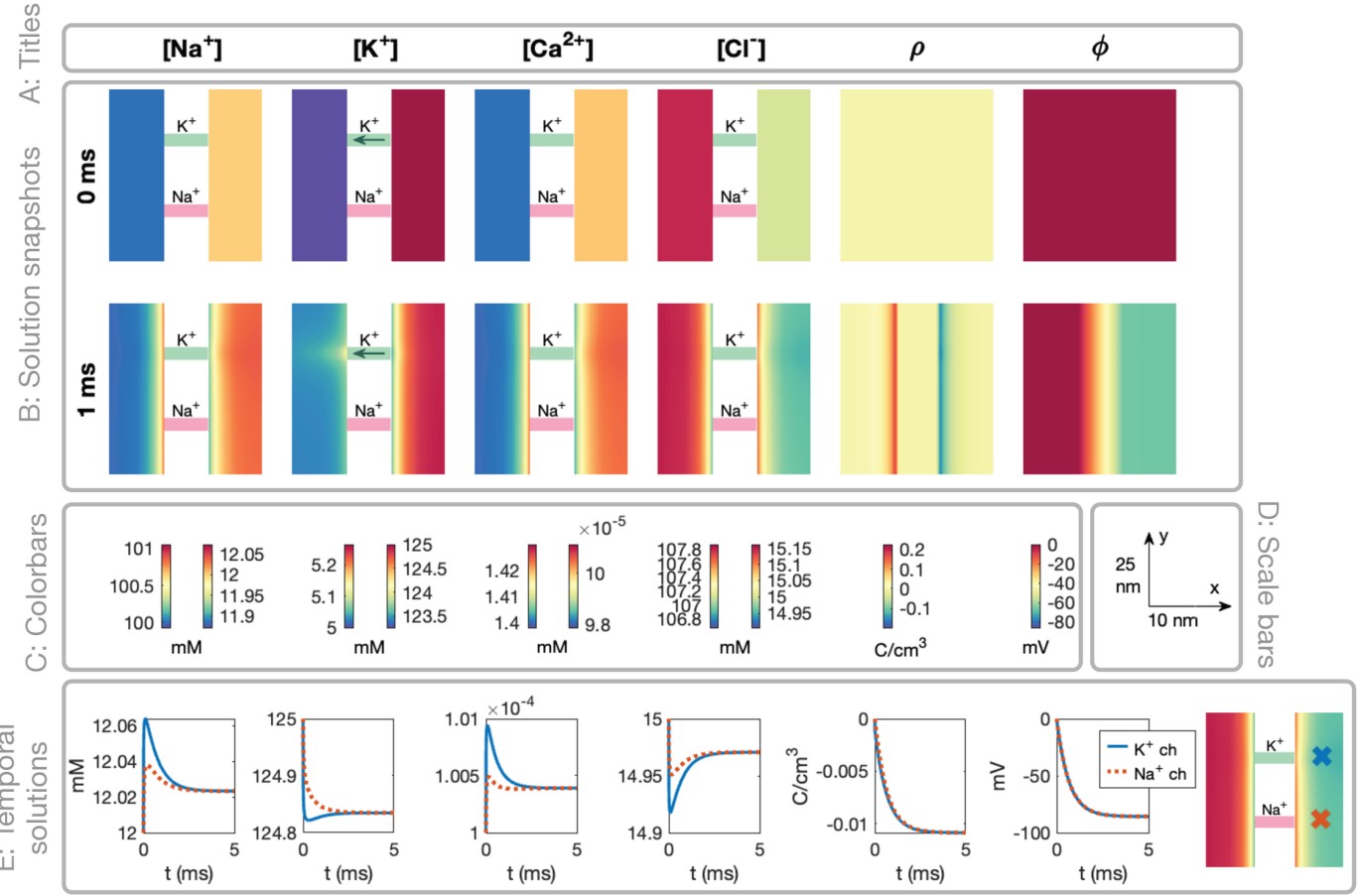

**Fig 6. Example illustration of the setup used to visualize the simulation results.** The setup is described in detail in Sect 3.3, and re-used in many figures below.

shows the colormap scaling in the intracellular space. The colorbar units are provided below the colorbars.

**D: Scale bars.** On the right-hand side of the colormaps, scalebars for the snapshots are provided for the $x$- and $y$-directions.

**E: Temporal solutions.** In the bottom row of the figures, the temporal evolution of the solutions in two spatial points are plotted. More specifically, we consider the points located in the center of the intracellular domain in the $x$-direction and at the location of two considered channels in the $y$- and $z$-directions. For the solutions displayed in Fig 6, this corresponds to the solutions 3.5 nm to the right of the $K^+$ and $Na^+$ channels. In the rightmost part of panel E, these two points are marked with crosses.

## 3.4. NP simulation of an open potassium channel. From electroneutrality to the cell's resting state.

After considering the simple examples in Figs 4 and 5, we will now consider simulations of the full three-dimensional (3D) setup described in the Methods section (see Fig 1). In these simulations, we include intracellular $Ca^{2+}$ binding buffer proteins, i.e., we solve the system (5)–(8).

We first consider the case when only a $K^+$ channel is open in the cell membrane. This simulation is used to obtain the resting state of the cell, used as initial conditions for the following simulations. The results of the simulation are displayed in Fig 7. In this visualization, we consider the solutions in a part of the domain that is close to the $K^+$ channel. Note that the extracellular part of the domain is depicted on the left-hand side and the intracellular part of the domain is depicted on the right-hand side. Details on the visualization setup are provided in Sect 3.3.

In Fig 7, we observe that, initially, all ionic concentrations are constant in the intracellular and extracellular parts of the domain (but different in these two parts of the domain). Moreover, the system fulfills electroneutrality ($\rho = 0$) and the electrical potential is zero everywhere. As the $K^+$ channel is opened, we observe that $K^+$ ions move from the intracellular to the extracellular part of the domain. This results in an elevated extracellular $K^+$ concentration and a reduced intracellular $K^+$ concentration, especially in the vicinity of the $K^+$ channel (see, e.g., $t = 0.1$ ms). As was observed in Fig 4, the resulting deviation from electroneutrality near the channel is counteracted by local changes in the other ionic concentrations. For example, we observe that the concentration of the positive ions ($Na^+$ and $Ca^{2+}$) increases slightly on the intracellular side of the $K^+$ channel, whereas the concentration of the negative $Cl^-$ ions oppositely decreases. These effects act to counteract the reduced $K^+$ concentration on the intracellular side of the channel.

Nevertheless, since no other ions than $K^+$ is able to cross the membrane, the movement of $K^+$ from the intracellular to the extracellular side of the membrane ultimately leads to a surplus of positive charges in the extracellular space and a surplus of negative charges in the intracellular space. As was also observed in Fig 5, this deviation from electroneutrality on opposite sides of the membrane results in a layer of positive charge density, $\rho$, on the extracellular side of the membrane and a layer of negative charge density on the intracellular side of the membrane (i.e., a Debye layer). For the positive ions ($Na^+$, $K^+$, and $Ca^{2+}$), this entails a layer of increased concentration on the extracellular side of the membrane and a layer of decreased concentration of the intracellular side. Conversely, a layer of decreased $Cl^-$ concentration forms on the extracellular side of the membrane and a layer of increased $Cl^-$ concentration forms on the intracellular side (see, e.g., $t = 5$ ms). Moreover, a potential difference is generated over the membrane. After a steady state solution is formed, the transmembrane potential, $v = \phi_i - \phi_e$, is about −80 mV, determined by the Nernst equilibrium potential of $K^+$ (see (13)).

In Fig 8, the solutions near the membrane at the end of the simulation is displayed, showing the details of the obtained Debye layer.

## 3.5. PNP simulation of an open sodium channel

Next, we consider the case of opening a $Na^+$ channel to generate an action potential upstroke. We start from the initial conditions representing the cell's resting state, taken from the end of the simulation displayed in Fig 7. We keep the $K^+$ channel open and open the $Na^+$ channel at $t = 0.05$ ms. Fig 9 shows the results of this simulation. We observe that after the $Na^+$ channel is opened, $Na^+$ moves from the extracellular to the intracellular space, resulting in a locally increased $Na^+$ concentration on the intracellular side of the $Na^+$ channel and a locally decreased $Na^+$ concentration on the extracellular side (see, e.g., $t = 0.11$ ms). To counteract the local deviation from electroneutrality, the concentration of the other ionic species also change locally in the vicinity of the $Na^+$ channel. For example, the $K^+$ and $Ca^{2+}$ concentrations decrease on the intracellular side of the $Na^+$ channel, and the $Cl^-$ concentration increases.

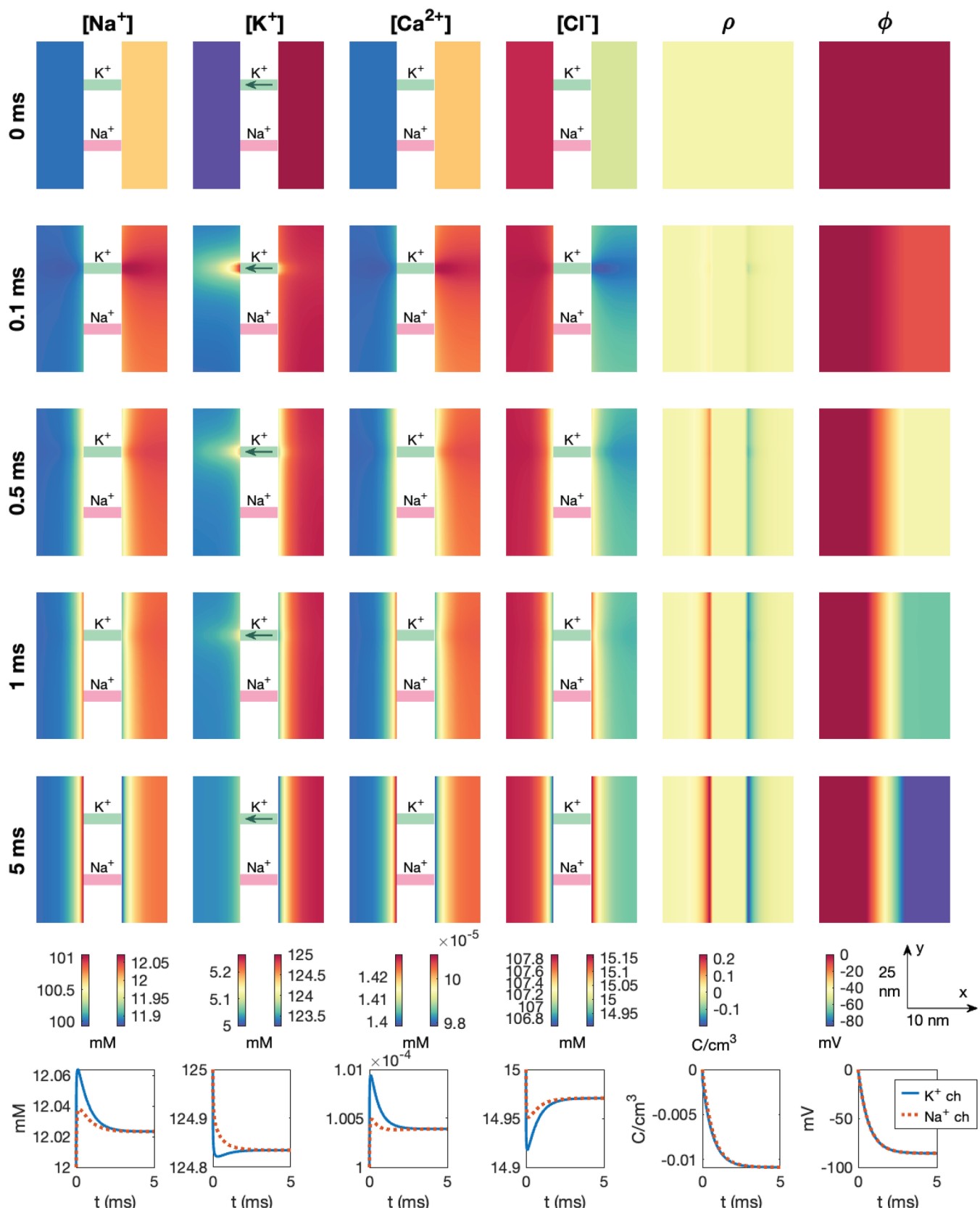

**Fig 7. Dynamics following the opening a K$^+$ channel in a PNP model simulation.** The figure setup is described in Sect 3.3. We use $\Delta t = 10\,\mu$s and an adaptive mesh like illustrated in Fig 3.

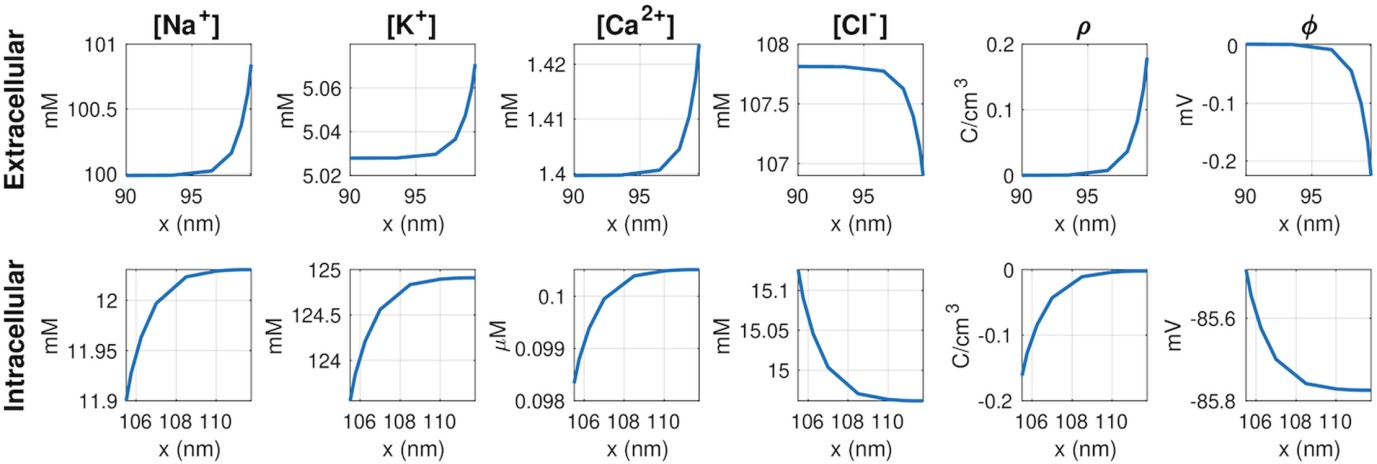

**Fig 8. Steady state solutions close to the membrane for the cell at rest.** Only a K$^+$ channel is open. The plots show the solutions along a line in the *x*-direction, taken from the simulation shown in Fig 7.

Moreover, the Na$^+$ ions moving from the extracellular to the intracellular domain, gradually reduce the surplus of positive charges in the extracellular space and the surplus of negative charges in the intracellular space, reducing the magnitude of $\rho$ near the membrane. In fact, eventually, there is a surplus of *negative* charges on the extracellular side of the membrane and a surplus of *positive* charges on the intracellular side (see $t = 0.5$ ms). The intracellular potential (and thus also the transmembrane potential) increases from about $-80$ mV to about 20 mV during the course of the simulation. This upstroke lasts for about 0.5 ms (see rightmost lower panel of Fig 9).

We also note that as the transmembrane potential is increased from the Nernst equilibrium potential for K$^+$, $v_{0,K^+}$, (see (15)) we also get current through the K$^+$ channel, which leads to local concentration changes close to the K$^+$ channel.

### 3.6. PNP simulation of an open calcium channel

After observing the dynamics following the opening of a K$^+$ channel leading to the resting state of the cell (Fig 7) and the opening of a Na$^+$ channel leading to the action potential upstroke (Fig 9), we will now consider the dynamics taking place in the dyad. We first consider the case of opening a Ca$^{2+}$ channel. More specifically, we start the simulation at the resting state with only a K$^+$ channel open. This K$^+$ channel remains open during the entire simulation. At $t = 0.05$ ms, the Na$^+$ channel is opened (like in Fig 9) starting the initial phase of the action potential upstroke. Subsequently, at $t = 0.1$ ms, we open the Ca$^{2+}$ channel in the dyad. The results of this simulation is shown in Fig 10, and in this case the visualization focuses on the area close to the Ca$^{2+}$ channel and NCX (i.e, the dyad).

We observe that a Debye layer is present in the dyad. Moreover, at $t = 0.09$ ms, following the Na$^+$ channel opening at $t = 0.05$ ms, the magnitude of $\rho$ near the membrane is reduced and the intracellular potential (and thus the transmembrane potential) is increased somewhat compared to the resting state ($t = 0$ ms). After the Ca$^{2+}$ channel is opened at $t = 0.1$ ms, the Ca$^{2+}$ concentration in the dyad increases, especially in the vicinity of the Ca$^{2+}$ channel. However, there is a clearly visible increase in the Ca$^{2+}$ concentration across the width of the dyad as well. In the lower panel, we observe that halfway across the dyad outside of the Ca$^{2+}$ channel, the Ca$^{2+}$ concentration increases from 0.0001 mM to about 0.7 mM. Further away in the

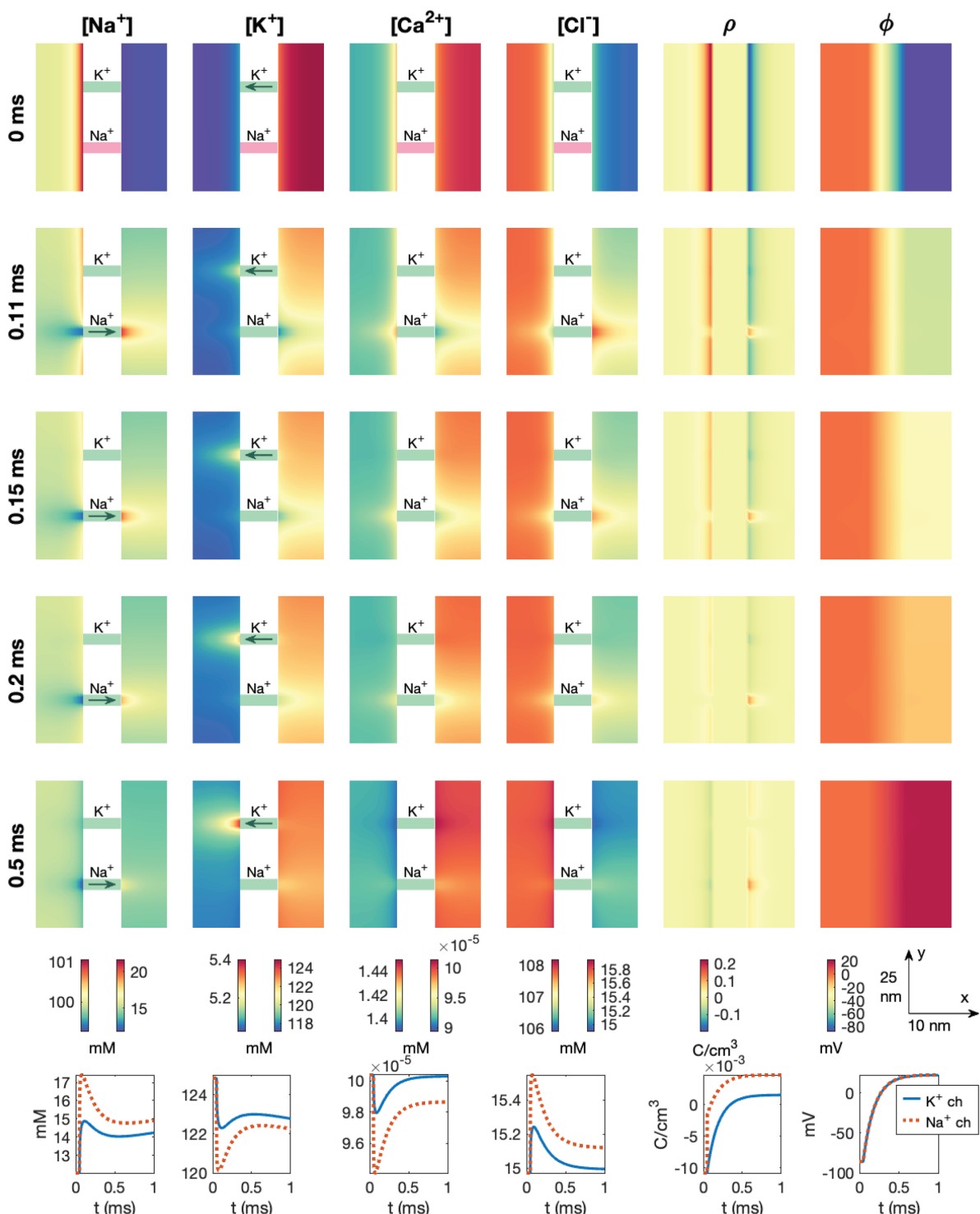

**Fig 9. Dynamics following the opening of a Na$^+$ channel in a PNP model simulation.** The figure setup is described in Sect 3.3. We use $\Delta t = 1\ \mu$s and an adaptive mesh like illustrated in Fig 3.

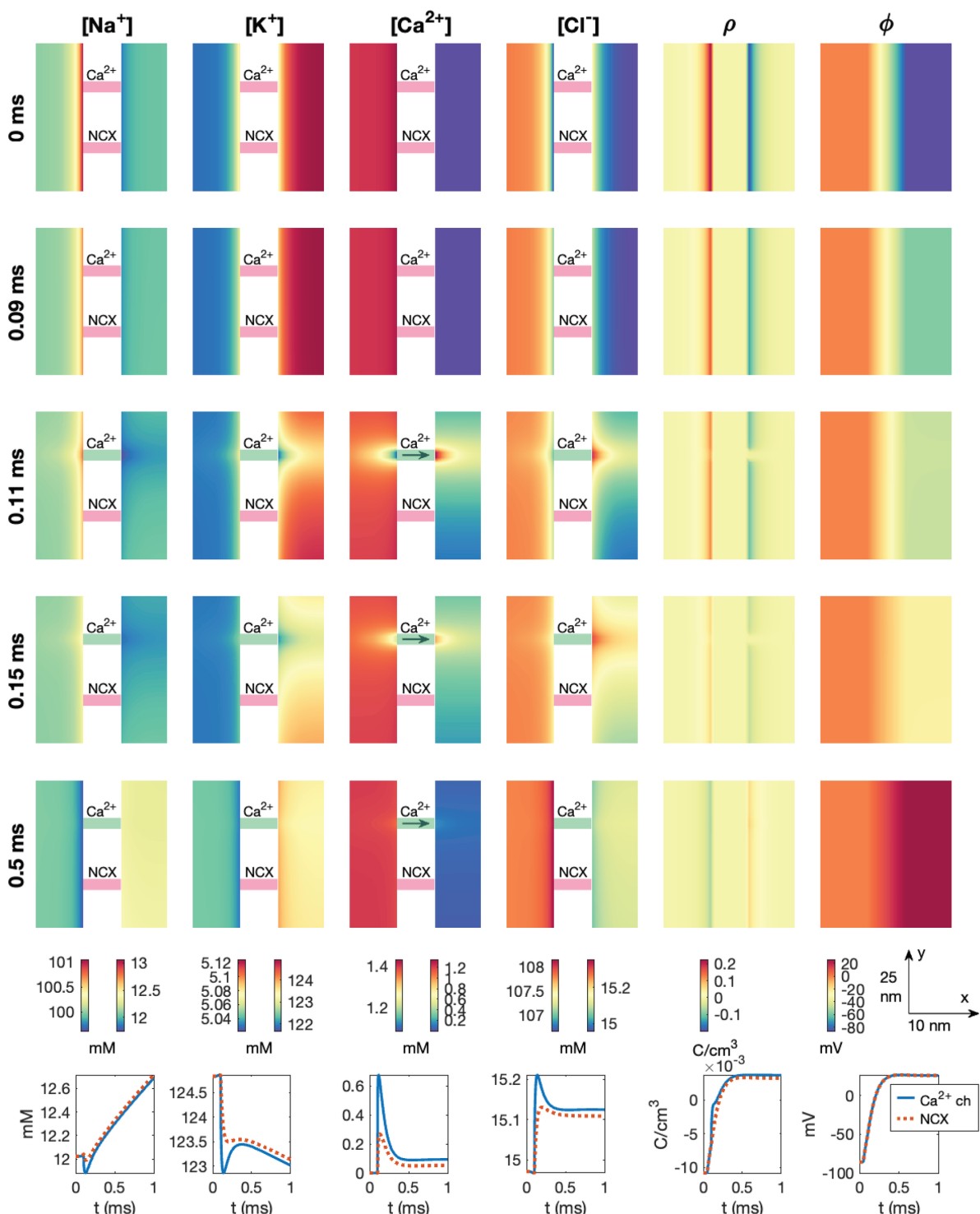

**Fig 10. Dyad dynamics following the opening of a Ca²⁺ channel in a PNP model simulation.** The figure setup is described in Sect 3.3. We use $\Delta t = 1\,\mu s$ and an adaptive mesh like illustrated in Fig 3.

$y$-direction, halfway across the dyad outside of the closed NCX, the concentration increases to about 0.3 mM. This increased $Ca^{2+}$ concentration could trigger RyR opening on the membrane of the SR. As observed in Figs 7 and 9, we also observe local changes in the remaining ionic species close to the $Ca^{2+}$ channel, counteracting the deviation from electroneutrality resulting from the $Ca^{2+}$ influx.

### 3.7. PNP simulation with a dyad including a sodium calcium exchanger and an open calcium channel

In Fig 11, we show the result of a simulation similar to the one displayed in Fig 10. The difference from Fig 10 is that the NCX is open, and it remains open during the entire simulation. Considering the arrows showing the direction of flow through the NCX, we observe that at rest, the NCX transports $Ca^{2+}$ out of the cell and $Na^+$ into the cell, but following the opening of the $Na^+$ channel, the flux changes direction (see $t = 0.09$ ms). After the $Ca^{2+}$ channel is opened, however, the direction changes back to transporting $Ca^{2+}$ out of the cell and $Na^+$ into the cell. Nonetheless, comparing the solutions of Figs 10 and 11, the presence of an open NCX does not appear to significantly change the dyad solutions. For instance, the $Ca^{2+}$ concentration in the points halfway across the dyad plotted in the lower figure panels appears to be very similar regardless of the presence of an open NCX.

### 3.8. PNP simulation of a dyad with a sodium calcium exchanger and no open calcium channel

To demonstrate a potential effect of an open NCX present in the dyad, we consider the case when the $Ca^{2+}$ channel is closed. The $K^+$ channel is open during the entire simulation, and the $Na^+$ channel is opened at $t = 0.05$ ms. The results of this simulation is displayed in Fig 12. We observe that in this case, the NCX flux remains in the direction that transports $Ca^{2+}$ ions into the cell after the $Na^+$ channel opening. At $t = 1$ ms, this results in a significant influx of $Ca^{2+}$ into the dyad through the NCX. Halfway across the dyad, the concentration reaches about $2 \cdot 10^{-4}$ mM $= 0.2$ $\mu$M. This increase in $Ca^{2+}$ concentration could potentially affect RyRs located in the apposing SR membrane.

### 3.9. PNP simulation with more prominent effects of the sodium calcium exchanger

To increase the effects of the NCX flux, we also consider a case in which the dyad width, $L_i$, is decreased to 5 nm, all intracellular diffusion coefficients are reduced to half of the default values reported in Table 1, and the $Na^+$ channel is moved to be in close proximity of the NCX. These changes are introduced in an attempt to increase the effect of the NCX flux in the dyad. We still keep the $Ca^{2+}$ channel closed. The results of this simulation are shown in Fig 13. We observe that the effect of the NCX flux is larger in this case, and the $Ca^{2+}$ concentration reaches a value of about 0.45 $\mu$M halfway across the dyad at $t = 1$ ms.

### 3.10. Investigating the effect of the dyad width and diffusion coefficient

In Fig 11, we presented the dyad dynamics in the default case of an open $Ca^{2+}$ channel and an NCX present in the dyad. We observed that following the $Ca^{2+}$ channel opening at $t = 0.1$ ms, the $Ca^{2+}$ concentration in the dyad increased. If RyRs were present at the right side of the dyad, this increased $Ca^{2+}$ concentration could have triggered the opening of RyRs on the membrane of the SR.

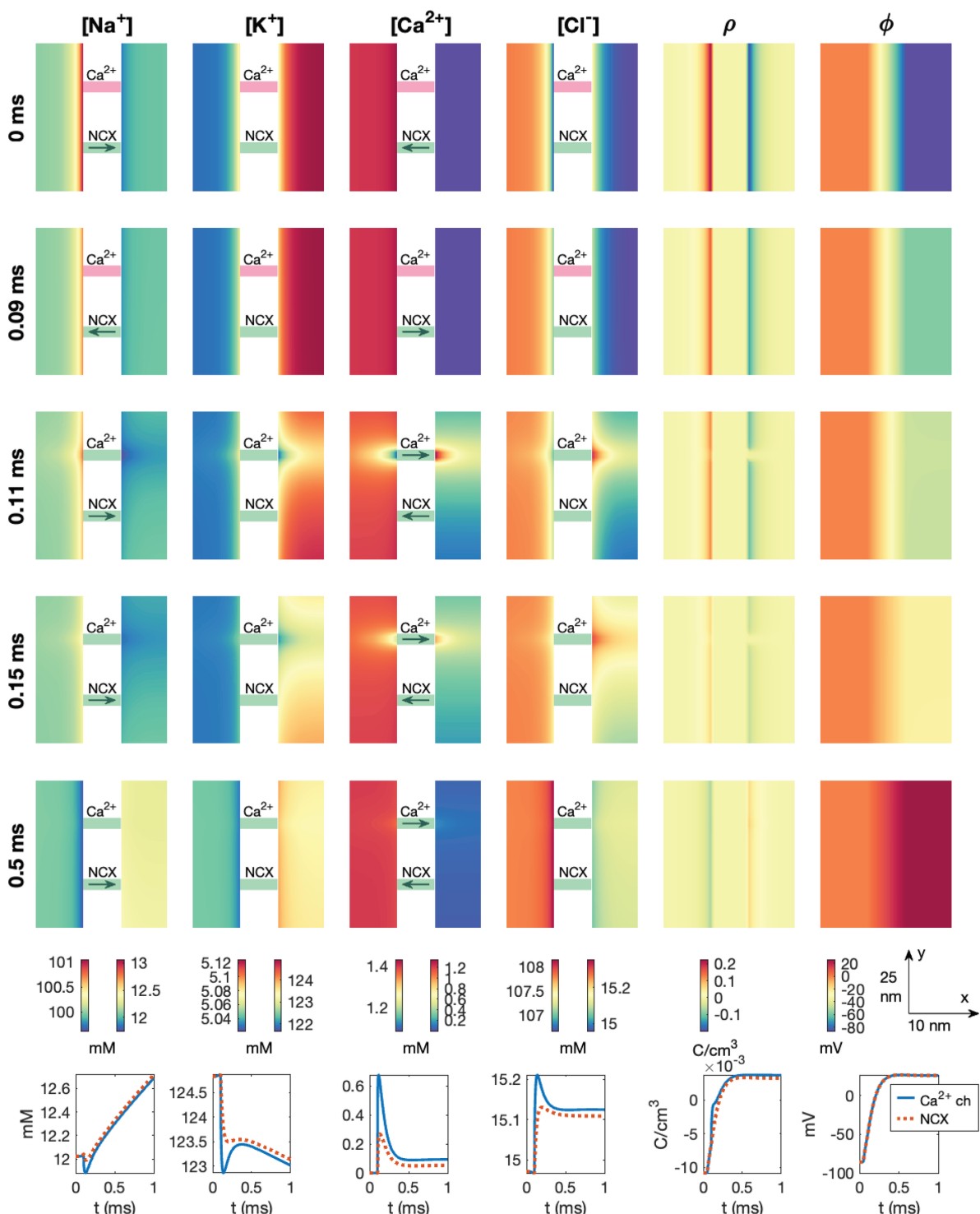

**Fig 11. Dyad dynamics following the opening of a Ca²⁺ channel in a PNP model simulation including an open NCX.** The figure setup is described in Sect 3.3. We use $\Delta t = 1 \,\mu$s and an adaptive mesh like illustrated in Fig 3.

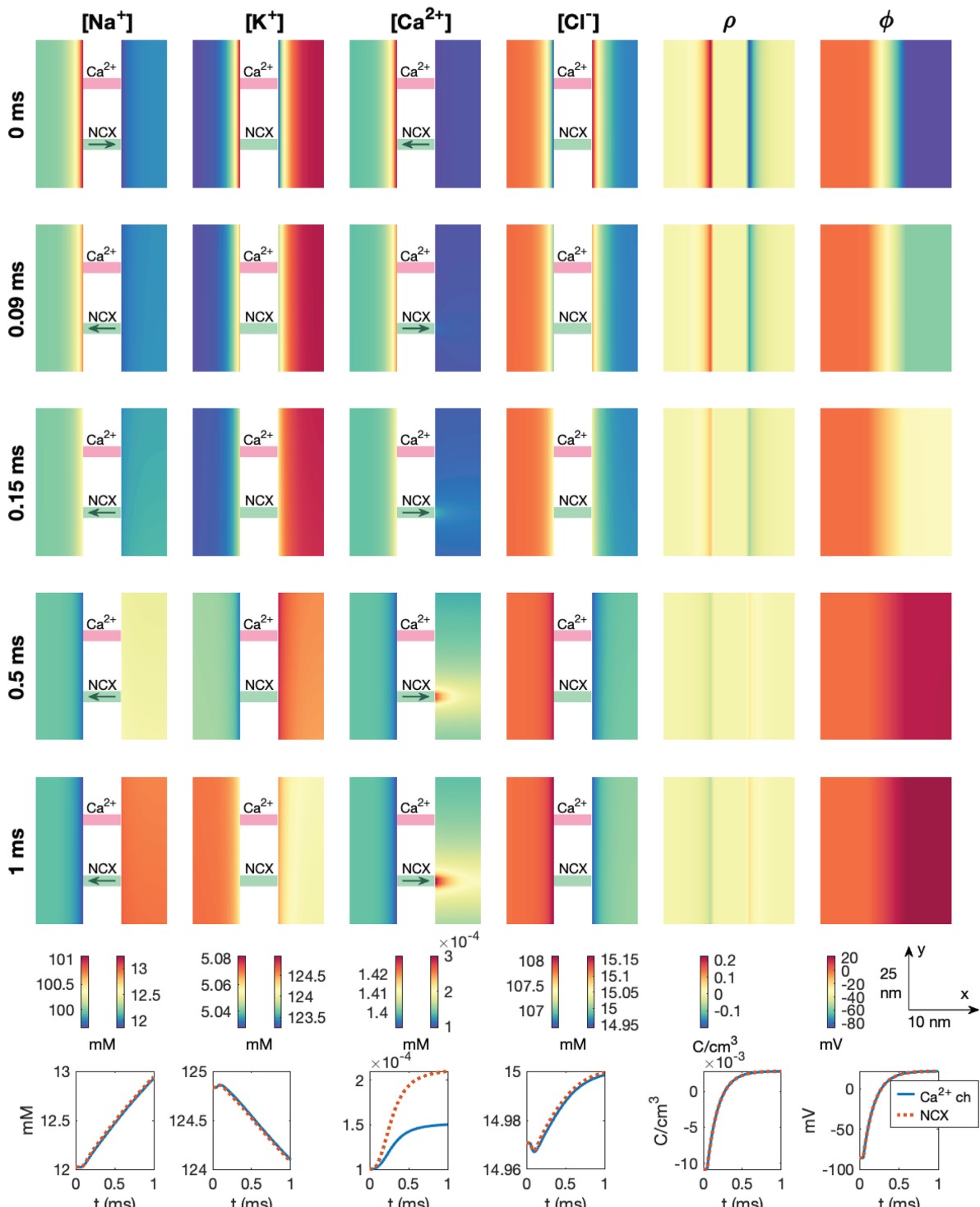

**Fig 12. Dyad dynamics following the opening of an Na$^+$ channel in a PNP model simulation including an NCX.** The Ca$^{2+}$ channel is not opened. The figure setup is described in Sect 3.3. We use $\Delta t = 1\,\mu$s and an adaptive mesh like illustrated in Fig 3.

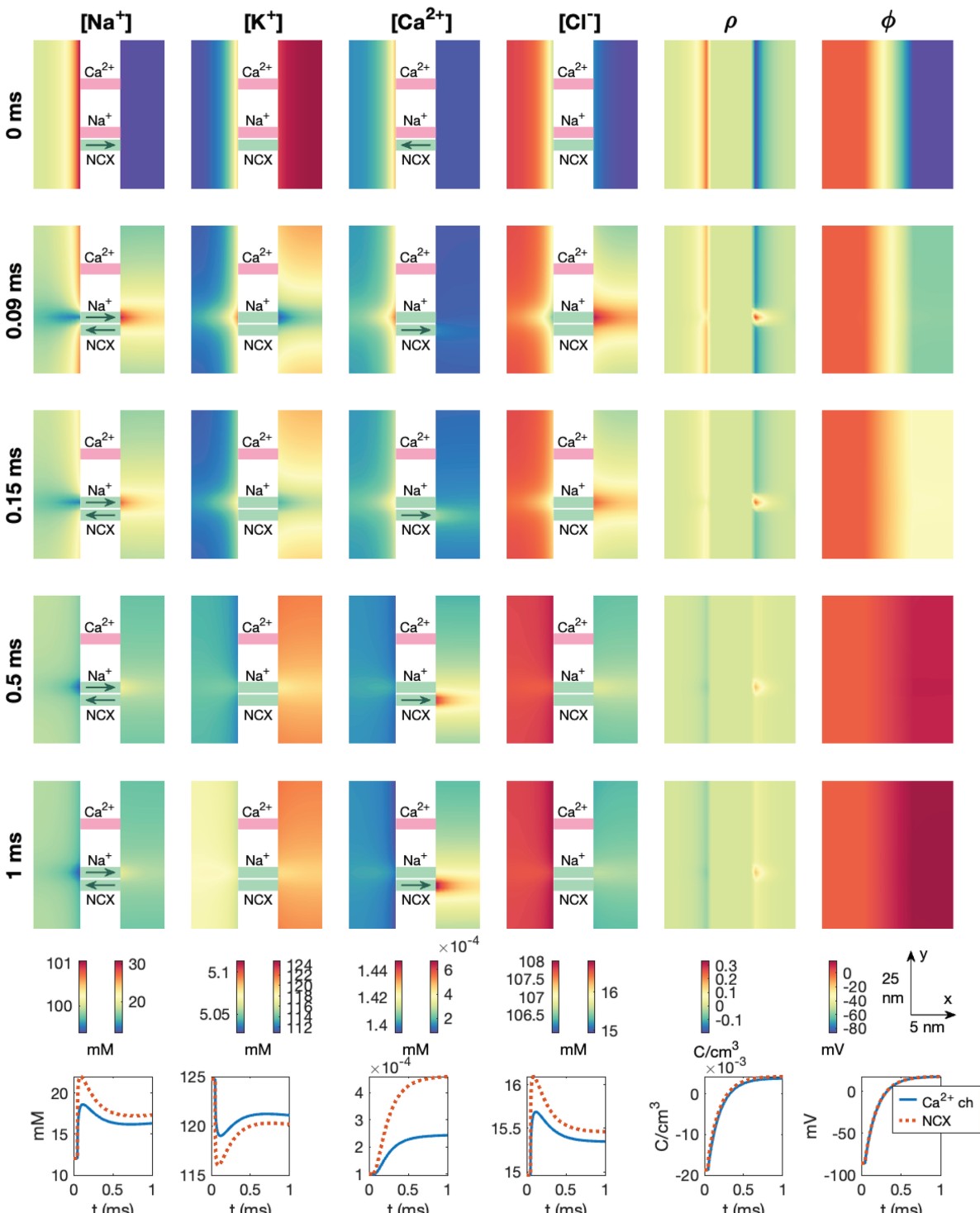

**Fig 13. Dyad dynamics following the opening of a Na$^+$ channel near an NCX in a PNP model simulation.** The Ca$^{2+}$ channel is not opened. The figure setup is described in Sect 3.3. We use $\Delta t = 1$ $\mu$s and an adaptive mesh like illustrated in Fig 3.

In Fig 14, we investigate how long it takes from the $Ca^{2+}$ channel is opened in the cell membrane until the RyRs would be triggered. We refer to this duration as the RyR activation time. The RyR activation times reported in Fig 14 are defined as the duration from the $Ca^{2+}$ channel is opened at $t = 0.1$ ms until the point in time when the average $Ca^{2+}$ concentration in an area corresponding to an RyR (i.e., an area spanning 30 nm × 30 nm directly across the dyad from the $Ca^{2+}$ channel) reaches a value above 0.5 $\mu$M, representing the threshold for RyR activation [36].

We investigate how the RyR activation time depends on the dyad width ($L_i$) and the intracellular $Ca^{2+}$ diffusion coefficient ($D_{Ca^{2+}}$). We consider dyad widths between 1 nm and 19 nm. Note that for dyad widths below 5 nm, we refine the resolution near the membrane from the default $\Delta x = 0.5$ nm to $\Delta x = 0.25$ nm. Additionally, we consider values of the intracellular $Ca^{2+}$ diffusion coefficient between 150 000 nm$^2$/ms and 300 000 nm$^2$/ms, based on different values applied for $Ca^{2+}$ diffusion in the dyad found in literature [30,39–43]. The intracellular diffusion coefficients for the remaining ionic species provided in Table 1 are scaled by the same factor as for $Ca^{2+}$.

Fig 14 shows that the RyR activation time increases in a non-linear manner as the dyad width is increased. Moreover, the RyR activation time increases as the diffusion coefficient is decreased.

## 3.11. Comparison to the pure diffusion model

We will now compare the above reported solutions of the PNP model to the solution of a pure diffusion or reaction-diffusion model.

**3.11.1. Decay following perturbations.** We start the comparison between the PNP and diffusion models by considering the simple example of an initial perturbation displayed in Fig 4. In Fig 15, we compare the solution of the full PNP system (1)–(3) to the solution of the pure diffusion equation

$$\frac{\partial c_k}{\partial t} = \nabla \cdot D_k \nabla c_k, \tag{39}$$

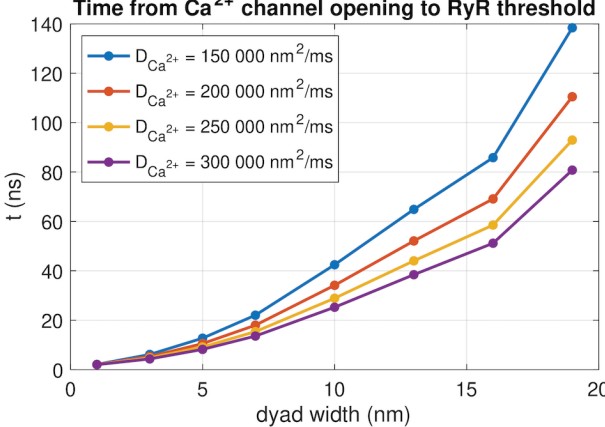

**Fig 14. RyR activation time for different values of the dyad width ($L_i$) and the intracellular $Ca^{2+}$ diffusion coefficient ($D_{Ca^{2+}}$).** The RyR activation time is defined as time from the membrane $Ca^{2+}$ channel is opened until the $Ca^{2+}$ concentration outside of an apposing RyR channel reaches 0.5 $\mu$M. We perform PNP model simulations similar to the one displayed in Fig 11. We use $\Delta t = 1$ ns and an adaptive mesh like illustrated in Fig 3.

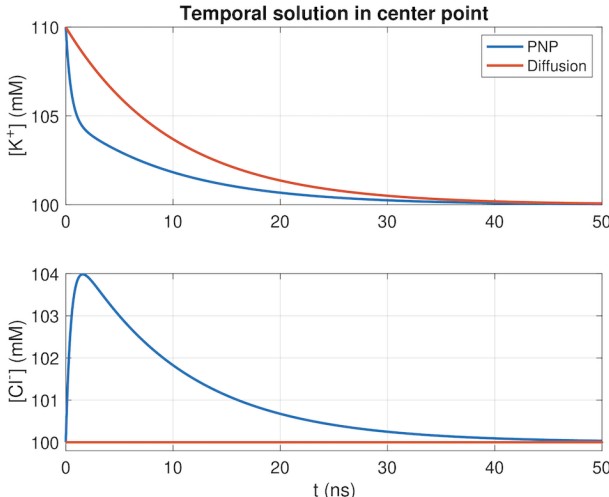

**Fig 15. Decay following a perturbation of one of the concentrations in simulations of the PNP model and the pure diffusion model.** We consider the simple 2D example with two ions displayed in Fig 4. The initial conditions of the two ions are shown in Fig 4A, and we plot the concentrations in the point in the center of the domain, like in Fig 4B. We have used $\Delta t$ = 0.1 ns and a uniform mesh with $\Delta x = \Delta y$ = 0.25 nm.

for $k$ = {K$^+$, Cl$^-$}. For the pure diffusion model, there is no equation terms driving the system towards electroneutrality. Therefore, the Cl$^-$ concentration is unaffected by the perturbation in the K$^+$ concentration and remains constant throughout the simulation. Furthermore, there are no initial fast dynamics driving the K$^+$ concentration towards electroneutality. Thus the decay of the K$^+$ concentration is slower for the pure diffusion model compared to the PNP model.

**3.11.2. Debye layer near a membrane.** We next consider the case of the Debye layer formation near a membrane investigated in Fig 5. Since the concentrations are constant on each side of the membrane and the membrane do not allow for diffusion, the right-hand side of the pure diffusion model (39) would be zero for the setup used as initial conditions for this example. Thus, the concentrations would remain constant, and no Debye layer would be formed in a pure diffusion model simulation. In other words, the PNP model is required to represent the Debye layer.

**3.11.3. Dyad dynamics.** We now consider the dyad simulation setup (see Fig 1C) and compare the solution of the full PNP system (5)–(8) to the solution of the reaction-diffusion equations

$$\frac{\partial c_k}{\partial t} = \nabla \cdot D_k \nabla c_k - \sum_{j \in B_k} J_{B_{k,j}}, \tag{40}$$

$$\frac{\partial b_{k,j}}{\partial t} = J_{B_{k,j}}, \tag{41}$$

where $J_{B_{k,j}}$ is defined in (4). We consider the same buffers and parameters as in the PNP model simulations (see Table 2).

In Fig 16, we show the solution of a reaction-diffusion model simulation of the same setup and open channels as in the PNP model simulation displayed in Fig 11. Since the reaction-diffusion system (40)–(41) do not give rise to an electrical potential, we have used the potential computed in the PNP model simulation to compute the transmembrane fluxes. Like in Fig 15, we observe that for the reaction-diffusion model a local increase in the $Ca^{2+}$ concentration do not affect any of the other ionic species like it does in the PNP model (compare Figs 11 and 16). In addition, no Debye layer is present for the reaction-diffusion model. Therefore, the concentrations of all ionic species except for $Ca^{2+}$ are constant in the dyad, unlike in the PNP simulation, where local gradients are present for all ions near the membrane. Nevertheless, the $Ca^{2+}$ concentration outside of the $Ca^{2+}$ channel, appears to be very similar in the reaction diffusion-model and the PNP model simulation (compare Figs 11 and 16).

In Fig 17, we investigate whether the differences between the PNP model and the reaction-diffusion model observed in Figs 11 and 16 has an effect of the time from $Ca^{2+}$ channel opening until the triggering of an RyR, defined as the RyR activation time. In Fig 17, we observe that the RyR activation time computed using the reaction-diffusion model appears to be virtually identical to that computed using the full PNP model. This also holds as the dyad width or the intracellular diffusion coefficients are adjusted. Fig 17 thus indicates that the reaction-diffusion model might be sufficient for capturing some of the important $Ca^{2+}$ dynamics in the dyad despite the solution differences from the PNP model observed in Figs 15 and 16.

In Fig 18, we investigate the similarity between the PNP model and the pure reaction-diffusion model by comparing the size of the three terms in (6),

$$B_d = \nabla \cdot D_k \nabla c_k, \tag{42}$$

$$B_e = \nabla \cdot \left( \frac{D_k z_k e}{k_B T} c_k \nabla \phi \right), \tag{43}$$

$$B_b = \sum_{j \in B_k} J_{B_{k,j}}, \tag{44}$$

for $k = Ca^{2+}$. We consider points at different distances, $d$, from the intracellular mouth of the $Ca^{2+}$ channel. We observe that the diffusion term, $B_d$, is by far the largest term in the equation. The small magnitude of the electrical term, $B_e$, compared to the diffusion term, $B_d$, explains why the difference between the PNP and the pure reaction-diffusion model is so small. In the Supplementary Information (see S3 Appendix), we similarly compare the size of the different terms for the $Na^+$ concentration following the opening of the $Na^+$ channel. We observe that for $Na^+$, the size of $B_e$ is more comparable to the size of $B_d$, and there is a more prominent difference between solutions of the PNP model and the pure reaction-diffusion model. Since the intracellular $Na^+$ concentration is about 1000 times larger than the intracellular $Ca^{2+}$ concentration in these simulations, this suggests that the negligible impact of the electrical term for the $Ca^{2+}$ concentration might be related to the small intracellular $Ca^{2+}$ concentration.

Note that a significant difference between the PNP model and the reaction-diffusion model is that the PNP model gives rise to an electrical potential whereas the reaction-diffusion model does not. If a reaction-diffusion model should be applied to study the $Ca^{2+}$ dynamics in the dyad, the transmembrane potential involved in the transmembrane fluxes would have to be computed in some other manner, e.g., by using a standard ODE model representation of the action potential, as, e.g., done in [36].

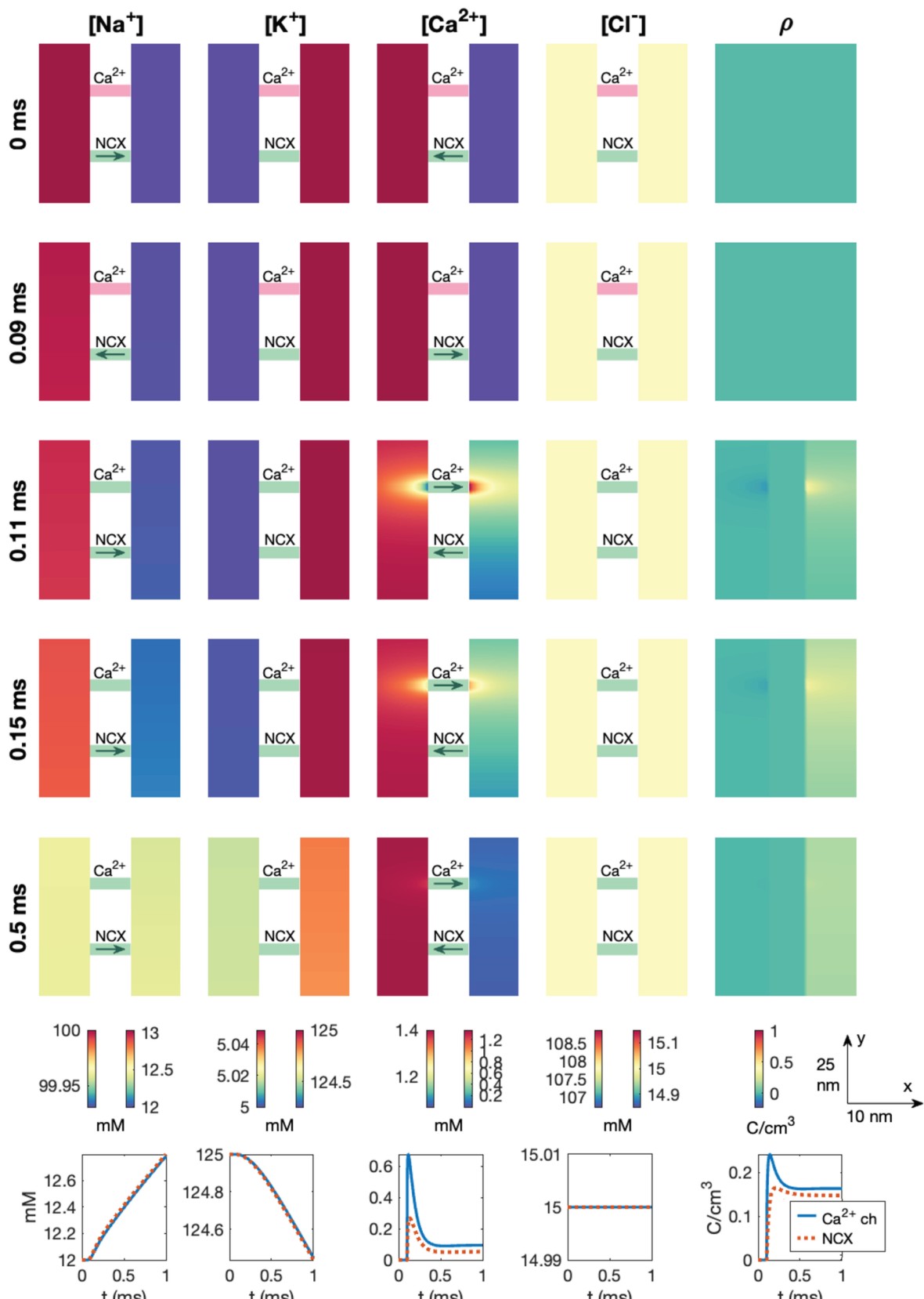

**Fig 16. Dyad dynamics following the opening of a Ca²⁺ channel in a pure reaction-diffusion model version of the PNP simulation displayed in Fig 11.** The figure setup is described in Sect 3.3. We have used $\Delta t = 1\ \mu$s and an adaptive mesh like illustrated in Fig 3.

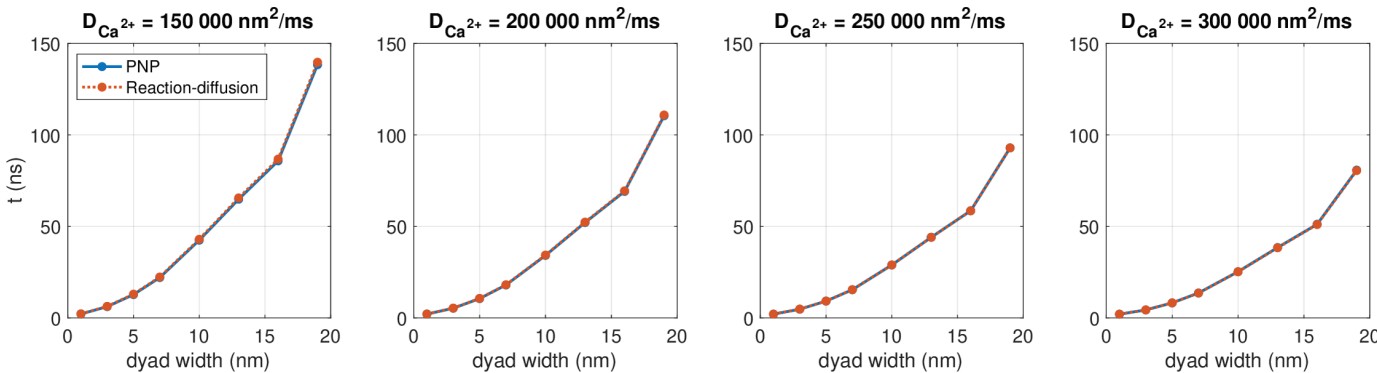

**Fig 17. RyR activation time for different values of the dyad width ($L_i$) and the intracellular Ca$^{2+}$ diffusion coefficient ($D_{Ca^{2+}}$) in the PNP and reaction-diffusion models.** The RyR activation time is defined as time from the membrane Ca$^{2+}$ channel is opened until the Ca$^{2+}$ concentration outside of an apposing RyR channel reaches 0.5 $\mu$M. The PNP results are also displayed in Fig 14. We have used $\Delta t$ = 1 ns and an adaptive mesh like illustrated in Fig 3. Note that the electrical potential computed in the PNP model is used to compute the transmembrane fluxes in the reaction-diffusion model.

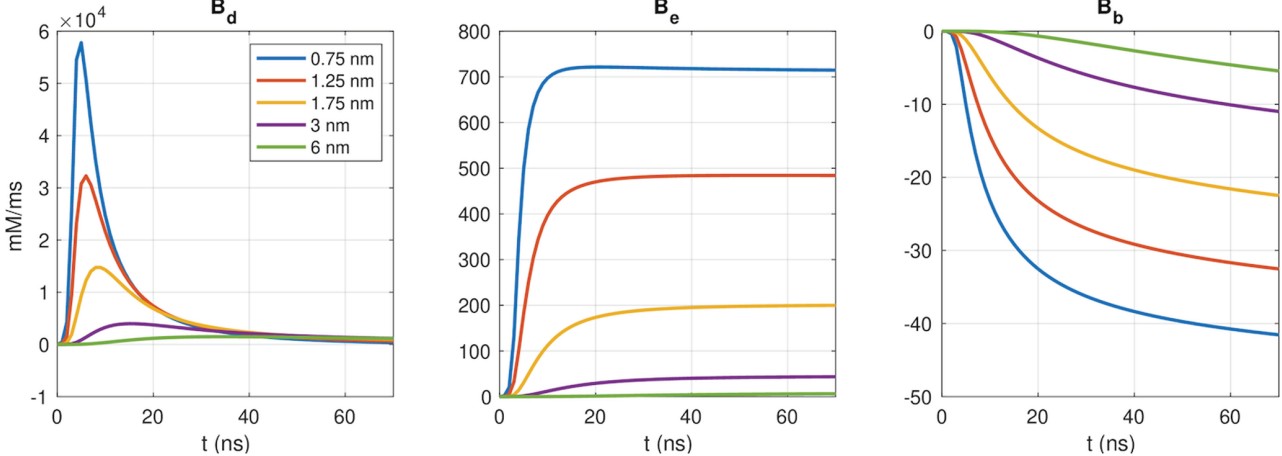

**Fig 18. Terms in the PNP model equation (6) governing the Ca$^{2+}$ concentration.** We consider the solution in the first 80 ns of simulation after the Ca$^{2+}$ channel is opened in points at different distances, $d$, in the $x$-direction from the intracellular mouth of the Ca$^{2+}$ channel. Note that the scaling of the $y$-axis is different in each panel. We have used the default model parameters, $\Delta t$ = 1 ns, and an adaptive mesh like illustrated in Fig 3.

## 3.12. Comparison to ODE model representations

We also wish to compare the PNP model of the dyad dynamics to a model that is based on further modeling simplifications. More specifically, we consider an ODE version of the dyad dynamics, similar to what is often applied in action potential models (see, e.g., [31,37,44,45]).

**3.12.1. ODE model representation with one dyad compartment.** We first consider an ODE model using the most typical representation of the dyad, i.e., by representing the dyad as one computational compartment, $\Omega_d$ (see Fig 19A). In addition, we consider one large surrounding cytosol compartment, $\Omega_l$. The concentrations in these two compartments are represented as constant in space, but varying in time, and we consider Ca$^{2+}$ ions and Ca$^{2+}$

**Fig 19. Illustration of two ODE model representations of the dyad dynamics.** The ionic concentrations are assumed to be constant (in space) in each considered compartment. A: The ODE model consists of one dyad compartment, $\Omega_d$, and one larger compartment representing the surrounding cytosol, $\Omega_l$. B: The ODE model consists of two dyad compartments (one near the $Ca^{2+}$ channel, $\Omega_c$, and one near the RyR, $\Omega_r$), in addition to the larger compartment representing the surrounding cytosol, $\Omega_l$.

binding buffers. $Ca^{2+}$ ions are allowed to diffuse between the two compartments, represented by the flux $J_{d,l}$. In addition, $Ca^{2+}$ flows into the dyad compartment through a membrane $Ca^{2+}$ channel ($J_{ch}$). The model equations of this ODE model are described in the Supplementary Information (see S4 Appendix).

In the leftmost panel of Fig 20, we have performed simulations using this ODE model to identify the RyR activation time, like for the PNP model in Fig 14. Since the dyadic $Ca^{2+}$ concentration, $c_d$, is assumed to be constant in space, this amounts to finding the time it takes from $Ca^{2+}$ channel opening until $c_d \geq 0.5\ \mu M$. The parameters of the ODE model are adjusted such that this time is the same as in the PNP model for $D_{Ca^{2+}} = 200\,000$ nm²/ms and $L_i = 7$ nm. We observe that as the dyad width is increased, the RyR activation time is increased, like also observed for the PNP model (see rightmost panel). However, the ODE model underestimates the increase in time as the dyad width in increased. In addition, altering the diffusion coefficient appears to have very little effect on the RyR activation time in this ODE model.

**3.12.2. ODE model representation with two dyad compartments.** In an attempt to more realistically capture the RyR activation time in an ODE model representation, we extend the ODE model illustrated in Fig 19A to include two different dyad compartments, with associated concentrations (see Fig 19B). One of the dyad compartments, $\Omega_c$, is set up to represent the half of the dyad closest to the membrane (and the $Ca^{2+}$ channel), and the other dyad compartment, $\Omega_r$, represents the half that is closest to the SR membrane (and the RyR). The two dyad compartments are connected by a diffusion flux, and the equations and parameters of the model are provided in the Supplementary Information (see S4 Appendix).

The center panel of Fig 20, shows the RyR activation times for the ODE model with two dyad compartments. We observe that for this model, the RyR activation time increases as the diffusion coefficient is reduced, like also observed for the PNP model. In addition, the increase in the RyR activation time increases more rapidly as a function of the dyad width compared to the one dyad compartment case. However, the two compartment ODE model still underestimates the the increase in time as the dyad width is increased compared to the

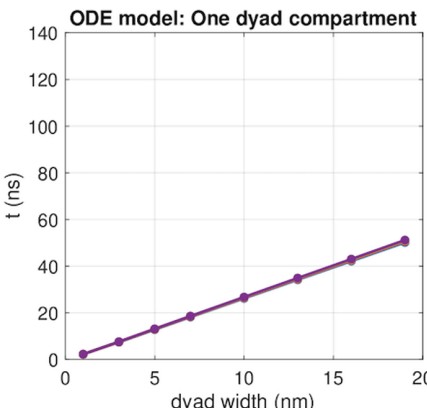 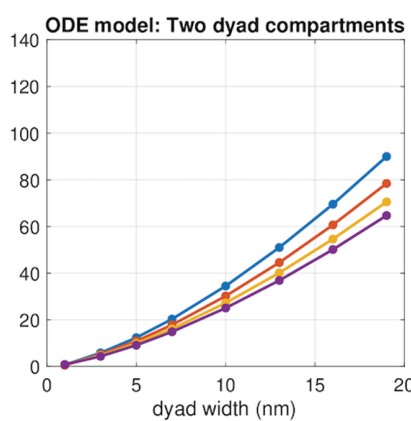 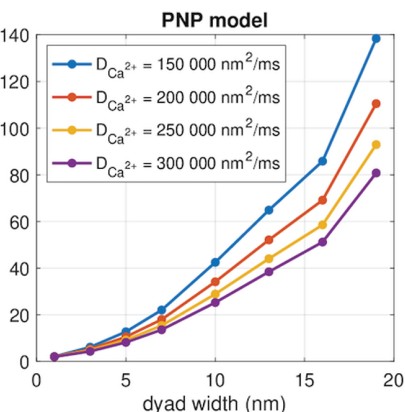

**Fig 20. Time from the Ca$^{2+}$ channel is opened until the [Ca$^{2+}$] outside of an apposing RyR channel reaches 0.5 $\mu$M in simulations of the two ODE models illustrated in Fig 19 and in the PNP model (left).** We have used $\Delta t = 1$ $\mu$s. Note that in the leftmost panel, the results for different values of $D_{Ca^{2+}}$ overlap.

PNP model (right panel). The results of Fig 20 thus suggest that a PNP model or a spatially resolved reaction-diffusion model may be required to accurately capture the details of the Ca$^{2+}$ dynamics in the dyad.

## 4. Discussion

### 4.1. Numerical solution of the PNP equations

Despite their ability to model essential biophysical processes, the PNP equations have received relatively little attention, primarily due to the significant computational cost associated with solving them. However, there is strong tradition for solving approximations of the system utilizing special geometries or symmetries or special features of the system, [46,47], and the full equations have also been solved, see, e.g., [25,26,38,48–51]. In [26], for instance, we had to use a time step of $\Delta t = 0.02$ ns to ensure numerical stability of the PNP system, making large-scale simulations impractical. In contrast, the numerical scheme employed in this study allows for a substantially larger time step of $\Delta t = 1000$ ns, enabling simulations of physiologically realistic upstroke times of approximately 0.5 ms. The key to this 50,000-fold increase in time step is solving the entire system in an implicit and fully coupled manner, as shown in (32) and (33), rather than splitting the potential equation (1) and electrodiffusion equations (2). This change in numerical approach allows us to select the time step based on the required accuracy, rather than being constrained by the severe stability restrictions of the previous method. In fact, with our original scheme, the simulations presented here would have been computationally prohibitive, requiring infeasible computing efforts – defined here as exceeding a week per simulation. The computations reported here have been performed on a modest computer (a Dell Precision 3640 Tower with an Intel Core processor (i9-10900K, 3.7 GHz/5.4 GHz) with ten cores with two threads), and the computing time for, e.g., Figs 9–13 was 6 hours for each simulation.

### 4.2. Electrodiffusion: What is the strongest term?

In Fig 4, we compare the magnitudes of the terms in the electrodiffusion equation (2) for a simple example of a perturbation from electroneutrality. The results indicate that electrodiffusion operates on two distinct time scales. Initially, the system rapidly establishes

electroneutrality, during which the electrical term

$$B_e = \nabla \cdot \left( \frac{D_k z_k e}{k_B T} c_k \nabla \phi \right) \tag{45}$$

dominates the dynamics. This phase is characterized by a swift redistribution of ions driven by electrostatic forces, which act to neutralize local charge imbalances. Once electroneutrality is achieved, the system enters a slower relaxation phase, where the decay toward chemical equilibrium is governed by the diffusive term

$$B_d = \nabla \cdot D_k \nabla c_k. \tag{46}$$

At this stage, the contribution of the electrical term $B_e$ becomes negligible, and the remaining dynamics are primarily dictated by standard diffusion. From a global perspective, $B_e$ is the strongest term when considering the entire simulation period. However, its dominance is transient; once electroneutrality is reached, the process becomes entirely diffusion-driven. This observation is in accordance with previous studies on electrodiffusion, see, e.g., [51], which emphasize the fundamental interplay between electrical forces and diffusion in neuronal environments and for other electrodiffusive processes.

In the simulations of intracellular dynamics following the opening of a membrane $Ca^{2+}$ channel, however, we observe that the electrical term, $B_e$, is almost negligible compared to the diffusion term, $B_d$ (see Fig 18). As a result, the pure reaction-diffusion equation was able to reproduce the time of arrival estimates for $Ca^{2+}$ at the RyR computed using the full PNP model (see Fig 17). The small magnitude of the $B_e$ term in the dyad likely reflects the low $Ca^{2+}$ concentration in that region (cf. S3 Appendix).

## 4.3. Modeling ion channels

The PNP model is often implicitly used to represent the ionic fluxes through channels in the cell membrane in mathematical models. Ion channel fluxes cannot be represented by only taking diffusion into account; without the electrical force, the huge gradients in the ionic concentrations across the membrane would simply diffuse through ion channels and the cell would be useless. In most models, the current through an ion channel is modeled by assuming one-dimensional flow across the membrane and then integrating the Nernst equation analytically, often approximated by a model of the form (15). It is also possible to regard the channel as part of the computational domain and solve the PNP equations in the entire volume; the extracellular space, the cell membrane including the channel and the intracellular space. In S1 Appendix we compare these approaches and find that the results are similar (as they should be), but not identical. The approach taken to represent the cross-membrane dynamics as part of the computational domain can be used for all the ion channels considered here, but not for the NCX. Modeling of the NCX has gained considerable attention and we have chosen to rely on the models in the literature. Since this flux does not follow directly from the PNP equations, we incorporate it using internal boundary conditions of the form (10) and (11). For consistency, we also treat the ion channel fluxes in the same manner.

## 4.4. Properties of electrodiffusion in the dyad

In Figs 10, 11, 12, and 13, we studied the ionic dynamics in the dyad when a $Ca^{2+}$ channel, an NCX, or both were open in the dyadic cell membrane. In Figs 10 and 11, we observed that when the $Ca^{2+}$ channel was open, the $Ca^{2+}$ concentration increased considerably close to the

$Ca^{2+}$ channel and this also resulted in local changes in the other ionic concentrations to maintain electroneutrality. Moreover, the $Ca^{2+}$ concentration increased across the dyad, potentially leading to the opening of RyRs on the membrane of the SR. Furthermore, we observed that the presence of an NCX in the dyad did not seem to significantly influence these dynamics. When the $Ca^{2+}$ channel was closed, on the other hand, we observed that $Ca^{2+}$ was transported into the dyad through the NCX. This resulted in a local increase in the dyadic $Ca^{2+}$ concentration outside of the NCX (see Fig 12). Moreover, if we reduced the intracellular diffusion coefficient and the dyad width and placed the $Na^+$ channel close to the NCX, the increased $Ca^{2+}$ concentration across the dyad approached the threshold concentration of 0.5 $\mu$M assumed to be required for RyR activation. Thus, under these conditions, an NCX could potentially trigger RyR opening in the absence of an open $Ca^{2+}$ channel. A similar result was found for a reaction-diffusion model in [36].

In Fig 14, we further examined the properties of electrodiffusion in the dyad by considering the time from membrane $Ca^{2+}$ channel opening until the concentration across the membrane (close to the RyRs) was high enough to trigger RyR opening. We observed that this time increased in a non-linear manner as the dyad width was increased. In addition, the time increased as the intracellular diffusion coefficient was decreased.

### 4.5. Is it necessary to solve the full PNP system?

As shown in Fig 17, the reaction-diffusion model (40)–(41) provides a time-of-arrival estimate for the $Ca^{2+}$-wave traveling from a calcium channel to the opposing RyR that is nearly indistinguishable from the result of the full PNP system (5)–(8), even though the two models (PNP and reaction-diffusion) exhibit some substantial differences close to the cell membrane (compare Figs 11 and 16). In particular, the reaction-diffusion model does not capture the altered concentration of other ionic species near the $Ca^{2+}$ channel or the altered ionic concentrations (including the $Ca^{2+}$ concentration) in the Debye layer. While it is theoretically possible to construct a scenario where a sustained electric potential gradient significantly influences ion transport even away from the cell membrane, such a setup would require carefully defined boundary conditions that may not be physiologically relevant. Based on our results, solving the reaction-diffusion equations appears sufficient for estimating time-of-arrival dynamics for the $Ca^{2+}$ concentration in the dyad. In Fig 20, we observe that even a simple ODE model with two compartments can roughly capture the effect of a changed diffusion coefficient or dyad width on the time of arrival, but not to the same degree of accuracy as the reaction-diffusion model. Moreover, a single-compartment model is insufficient to reproduce the effect of a changed diffusion coefficient.

### 4.6. Simplifications in the representation of the dyad

In this study, we have applied the PNP equations to model ionic dynamics in the dyad. The PNP model has the potential of providing a more detailed and accurate representation of the electrodiffusion dynamics than the more commonly applied reaction-diffusion models. Our focus has been to use this detailed model for the dynamics in an otherwise relatively simple setup. Thus, simplifications have been applied to other aspects of the modeling representation. For example, we have only focused on the dynamics taking place before the activation of RyRs in the membrane of the SR. Therefore, the model used in this study does not include a representation of the SR or RyRs, but this would be a natural extension of the current framework.

In addition, the present model uses a simplified rectangular geometry for the dyadic space and assumes deterministic gating of single ion channels. These choices abstract away much

of the structural complexity and stochastic channel behavior that are known features of the cardiac dyad. In particular, detailed reconstructions [30,52] have shown that the surfaces of the dyadic cleft and the SR membrane can be highly irregular, potentially leading to spatial heterogeneities in ion concentrations and electric fields that are not captured by the present approach. Also, ion channels in physiological membranes often appear in clusters rather than as isolated entities, see, e.g., [53,54], but for simplicity, we have used only one channel of each type in our simulations. This entails one $Na^+$ channel per 1 $\mu m^2$, tuned to reproduce a realistic upstroke velocity within the modeled domain. Furthermore, experimental studies and reviews [55] have demonstrated that clusters of L-type $Ca^{2+}$ channels and RyRs display stochastic gating behavior and possible coupled gating phenomena. Our model does not include these aspects, as we chose to focus on the electrodiffusion dynamics under the assumption of deterministic channel behavior. This was done to isolate and examine the fundamental consequences of electrical forces on ionic transport without the additional complexity of geometry and gating stochasticity. In this study, we have also only included two types of stationary $Ca^{2+}$ binding buffers in the dyad, but other types of buffers, including mobile buffers could also be included (see, e.g., [30]). Another simplification in the present approach is that also the part of the domain that does not represent the dyad area (see, e.g., Fig 3) is represented by the same narrow intracellular space and the same $Ca^{2+}$ binding proteins as those present in the dyad. A wider intracellular space and different $Ca^{2+}$ binding proteins in this part of the domain would probably be more realistic.

## 4.7. More advanced computational models of electrodiffusion dynamics

Several more detailed modeling approaches exist that extend beyond the standard PNP formalism used here. These include modified electrodiffusion models that incorporate finite ion sizes (elaborate models of steric effects), ion-ion correlations, or additional corrections at the nanoscale, see, e.g., [56]. In the Supplementary Information (see S5 Appendix), we extend the PNP model described above to account for steric effects following the approach in [56]. However, we observe that including steric effects do not have a significant effect on the results under the conditions considered in this study.

Alternative approaches to modeling electrodiffusion at the nanometer scale include models that describe ion transport directly within the channel structures themselves, using PNP equations with steric effects or related formalisms, see [57]. These models address ion interactions and crowding within the confined environment of the channel pore, representing a more accurate and elaborate modeling strategy to the approach taken here.

## 4.8. Continuous vs. discrete modeling of the dyad

As mentioned above, the PNP system serves as a continuous model for electrodiffusion in the dyad. However, the dyadic volume is extremely small, and ionic concentrations – particularly for $Ca^{2+}$ – are very low; in fact, the number of $Ca^{2+}$ ions is often below one at any given instant, [58]. The average volume of the dyad has been measured to be $4.39 \times 10^5$ $nm^3$, [59], which is approximately $4 \times 10^7$ times smaller than the volume of an adult cardiomyocyte (16 pL), [60]. To accurately simulate this volume, we employ a spatial resolution at the nanometer scale and a temporal resolution at the nanosecond scale.

Mathematically, the PNP equations remain well-defined even at very low concentrations, but their physical interpretation becomes less straightforward. In [58], it is argued that continuous models should be interpreted as time-averages of inherently stochastic processes. An alternative approach is to use particle-based stochastic models, [61], which explicitly represent individual ion movements. Such stochastic models are frequently used to study

$Ca^{2+}$ dynamics in small, confined spaces where ion numbers are low and concentration-based descriptions may fail, [62–64]. Comparisons between stochastic and deterministic models generally indicate that for sufficiently small volumes, stochastic discrete models are more accurate than standard reaction-diffusion descriptions, particularly when dealing with nanometer-scale $Ca^{2+}$ domains, [58,65]. However, these comparisons are typically performed against reaction-diffusion equations or pure diffusion models, not against the PNP system itself. This distinction is critical because reaction-diffusion models, unlike PNP, do not account for the electrical forces governing ion movement in confined spaces. The presence of local charge imbalances and electric fields at the nanometer scale may significantly influence ion dynamics in ways that neither stochastic nor reaction-diffusion models capture.

Our approach is to analyze the PNP equations while acknowledging that they represent a continuous approximation of inherently discrete events. The results must therefore be interpreted with the caveat that local charge densities do not necessarily reflect instantaneous ion positions but rather their expected distributions over time.

## 5. Conclusion

In this study, we present a nano-scale computational model of electrodiffusion in the dyadic space using the Poisson-Nernst-Planck (PNP) equations. Our results confirm that resolving electrodiffusion dynamics at the nanometer scale provides critical insights into ionic transport mechanisms. We applied an improved numerical scheme for the PNP equations, which enables longer time steps and longer simulations, while maintaining numerical stability. This allowed us to investigate key factors influencing $Ca^{2+}$ arrival at the ryanodine receptors, showing that both the diffusion coefficient and the dyad width significantly impact the time-of-arrival dynamics in a non-linear manner. Furthermore, we demonstrated that the sodium-calcium exchanger may trigger ryanodine receptor activation in the absence of an open $Ca^{2+}$ channel if the dyad is narrow, the intracellular diffusion coefficient is reduced, and a $Na^+$ channel is present in the dyad. Our findings also confirm the formation of a Debye layer near the cell membrane, which cannot be reproduced without including the electrical terms in the PNP equations. Finally, we highlighted that cross-species ionic interactions in the dyad are purely electrical, meaning that they do not manifest in pure reaction-diffusion models. These results underscore the necessity of nano-scale modeling for accurately capturing ionic dynamics in the dyad. The choice of model depends on the specific application: if representation of the Debye layer and cross-species ionic interactions are required, the full PNP system should be applied. However, if this is not a concern, a much simpler reaction-diffusion model could be an adequate approximation.

## Supporting information

**S1 Fig. Dynamics following the opening of a $Ca^{2+}$ channel in a PNP model simulation without $Ca^{2+}$ binding buffers.** The simulation is the same as that displayed in Fig 12 in the main paper, except that there are no $Ca^{2+}$ binding buffers present.
(PDF)

**S2 Fig. $Ca^{2+}$ concentration 3.5 nm to the left of the $Ca^{2+}$ channel and the NCX in simulations with and without $Ca^{2+}$ binding buffers present.** The simulation with $Ca^{2+}$ buffers is displayed in Fig 12 in the main paper and the simulation without $Ca^{2+}$ buffers is displayed in S1 Fig.
(PDF)

**S1 Appendix. Representing ion channels as membrane pores.**
(PDF)

**S2 Appendix. Convergence of the numerical scheme for the PNP model.**
(PDF)

**S3 Appendix. Comparison of equation term sizes for $Na^+$.**
(PDF)

**S4 Appendix. ODE model representations.**
(PDF)

**S5 Appendix. Dyad dynamics in a PNP model incorporating steric effects.**
(PDF)

## Disclosure of writing assistance

During the preparation of this manuscript, the authors utilized the ChatGPT4o language model to enhance the language quality for contributions from non-native English speakers. Subsequent to this automated assistance, the authors rigorously reviewed and edited the manuscript to ensure its accuracy and integrity. The authors assume full responsibility for the content of the publication.

## Author contributions

**Conceptualization:** Karoline Horgmo Jæger, Aslak Tveito.

**Data curation:** Karoline Horgmo Jæger.

**Formal analysis:** Karoline Horgmo Jæger, Aslak Tveito.

**Funding acquisition:** Aslak Tveito.

**Investigation:** Karoline Horgmo Jæger, Aslak Tveito.

**Methodology:** Karoline Horgmo Jæger, Aslak Tveito.

**Project administration:** Karoline Horgmo Jæger, Aslak Tveito.

**Software:** Karoline Horgmo Jæger.

**Visualization:** Karoline Horgmo Jæger.

**Writing – original draft:** Karoline Horgmo Jæger, Aslak Tveito.

**Writing – review & editing:** Karoline Horgmo Jæger, Aslak Tveito.

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
