## [Decision Letter · Decision Letter 0]

21 Apr 2025

PCOMPBIOL-D-25-00509

Electrodiffusion dynamics in the cardiomyocyte dyad at nano-scale resolution using the Poisson-Nernst-Planck (PNP) equations

PLOS Computational Biology

Dear Dr. Jæger,

Thank you for submitting your manuscript to PLOS Computational Biology. After careful consideration, we feel that it has merit but does not fully meet PLOS Computational Biology's publication criteria as it currently stands. Therefore, we invite you to submit a minor revision of the manuscript that addresses the points raised during the review process.

Please submit your revised manuscript within 30 days Jun 21 2025 11:59PM. If you will need more time than this to complete your revisions, please reply to this message or contact the journal office at ploscompbiol@plos.org. Please include the following items when submitting your revised manuscript:

We look forward to receiving your revised manuscript.

Kind regards,

Christopher E Miles

Academic Editor

PLOS Computational Biology

Feilim Mac Gabhann

Editor-in-Chief

PLOS Computational Biology

**Journal Requirements:**

**Reviewers' comments:**

Reviewer's Responses to Questions

**Comments to the Authors:**

Reviewer #1: In the present paper, the authors study electrodiffusion dynamics near the cell membrane of a cardiac cell. To model the electrical properties of cardiac cells, normally either average concentrations, or spatial concentrations following reaction-diffusion equations have been considered in the literature. A pertinent question, that the authors try to answer in this paper (and that I have also asked myself) is whether electrical effects are important at short spatial and/or temporal scales. To study these effects, the authors use Poisson-Nernst-Planck (PNP) equations, and compare their results with those obtained using a pure reaction-diffusion scheme, or a model of compartments. Quite reassuring, they find that, although a Debye layer is formed near the surface, the reaction-diffusion equations are able to describe correct many phenomena of interest.

As I said, I find this a very relevant paper to understand the limits of the models normally used for describing the electrical properties of cardiac cells. This study has also its limits (use of a continuous description, stochastic dynamics of the ion channels, phenomenological buffers, etc). The authors are well aware of these limitations and discuss them in the final section of the paper. Maybe another aspect that the authors could include in the discussion is the number of channels in the model. If I understand correctly, the authors consider an area of the cell membrane of 1 microm^2. In this area there should be a decent number of channels. In fact, some channels come often in clusters. The authors should discuss this point and whether it could affect their results.

Reviewer #2: See attachment.

Reviewer #3: This manuscript presents a computational investigation into the electrodiffusion dynamics of ions within the cardiac dyad, employing the Poisson-Nernst-Planck (PNP) system of equations. The authors developed a three-dimensional computational model based on the PNP equations, simulating the concentrations of sodium (Na+), potassium (K+), calcium (Ca2+), and chloride (Cl−) ions, along with the electrical potential (ϕ). The model geometry consists of a simplified rectangular representation of the dyadic cleft (intracellular space), a segment of the adjacent cell membrane incorporating single, deterministically gated K+, Na+, and Ca2+ channels and the Na+/Ca2+ exchanger (NCX), and a portion of the T-tubule (extracellular space). The model also includes two types of stationary intracellular Ca2+ buffers, acknowledging their significant role in modulating free Ca2+ concentration. The PNP system is solved numerically using a finite difference method on an adaptive mesh, employing operator splitting to handle buffering reactions and an implicit time-stepping scheme for the coupled electrodiffusion equations.

The study reports several key findings. Simulations demonstrate the formation of a Debye layer—a region of non-electroneutrality—near the cell membrane at rest, emphasizing the contribution of electrical forces even in establishing equilibrium within the dyad. The results suggest that interactions between different ion species within the dyad are primarily mediated by electrical forces, an effect inherently missed by models based solely on diffusion. A sensitivity analysis reveals that the local Ca2+ concentration profile and the time required for Ca2+ ions released near the membrane to reach the vicinity of ryanodine receptors (RyRs) on the SR (used as a proxy for RyR activation time) are significantly dependent on the assumed width of the dyadic cleft and the intracellular Ca2+ diffusion coefficient (DCa). Furthermore, the simulations explore a scenario where localized Na+ accumulation near an open Na+ channel could potentially drive the NCX to operate in reverse mode sufficiently to trigger RyR opening, even without direct Ca2+ influx through L-type calcium channels (LTCCs). Comparisons with simpler models indicate that while the PNP model predicts faster relaxation towards electroneutrality compared to pure diffusion, a reaction-diffusion model yields similar estimates for the Ca2+ arrival time at RyRs, despite differences in the predicted ion concentrations very close to the membrane. ODE-based compartmental models failed to reproduce the sensitivity to dyad width and DCa observed in the PNP simulations.

The authors conclude that the necessity of employing the full PNP system for modeling the cardiac dyad depends critically on the specific scientific question being addressed. They propose that PNP is essential when investigating phenomena directly related to the Debye layer or electrical interactions between different ion species. However, for estimating the timing of Ca2+ signal propagation across the dyad to the RyRs, simpler reaction-diffusion models might provide adequate accuracy

The findings of this paper offer potentially useful perspectives on the interplay between diffusive transport and electrical forces within this critical signaling domain. The novelty rests primarily on the specific application and the comparative aspect, rather than on methodological innovation. The work is a valuable contribution which applies a fundamental electrodiffusion model to the cardiac dyad and performs comparative analyses.

Nonetheless, the paper could benefit from a revisions according to the points below, in order to rigorously address its limitations, particularly the highly simplified geometry used, the deterministic gating for single ion channels which do not take into account the known structural complexity and stochastic behavior of the dyad, and the known limitations of the standard PNP formalism itself, particularly concerning its applicability in nanoscale environments where effects like finite ion size and ion-ion correlations, neglected in the standard model, can become prominent.

Major Points:

1. Justification and discussion of Standard PNP: The authors should provide a clear rationale for employing the standard PNP formulation in the nanoscale dyadic environment, explicitly addressing why potential limitations related to ion size (steric effects) and ion-ion correlations are considered negligible or acceptable for the study's goals; see e.g. Zheng, Qiong, and Guo-Wei Wei. "Poisson–Boltzmann–Nernst–Planck model." The Journal of chemical physics 134.19 (2011). This discussion should consider the potential impact of these neglected physical factors on the predicted concentration profiles, Debye layer structure, and electrical interactions. The limitations section should explicitly mention these standard PNP assumptions, and conclusions should be tempered accordingly.

2. Impact of geometry simplifications: The manuscript could benefit from an expanded discussion on the consequences of using a simplified rectangular geometry and deterministic single-channel gating. How might realistic, irregular dyad morphologies and the stochastic, clustered nature of LTCC and RyR gating (see e.g. Jones, Peter P., Niall MacQuaide, and William E. Louch. "Dyadic plasticity in cardiomyocytes." Frontiers in Physiology 9 (2018): 1773) alter the reported findings concerning Debye layers, cross-species electrical interactions, sensitivity to parameters, and the NCX-triggering scenario?

3. Deeper interpretation of PNP vs. RD comparison: The observation that PNP and RD yield similar Ca2+ arrival times could benefit from a deeper analysis. The authors should investigate and discuss the reasons behind this similarity. Does it imply electrical forces are truly negligible for bulk transport across the cleft under these conditions, or are other factors at play? This analysis is crucial for substantiating the conclusion about the conditional sufficiency of RD models.

4. Contextualization with advanced models: The discussion should briefly acknowledge the existence of more sophisticated electrodiffusion models (e.g., PNPF, PNPB, steric PNP) and advanced stochastic dyad models to provide better context for the current work's position, scope, and limitations within the broader field of computational dyad modeling (see e.g. Colman, Michael A., et al. "Multi-scale computational modeling of spatial calcium handling from nanodomain to whole-heart: Overview and perspectives." Frontiers in Physiology 13 (2022): 836622).

5. Numerical validation evidence: In order to bolster confidence in the simulation results, it would be interesting to provide evidence of numerical convergence (e.g., mesh refinement study showing convergence of key outputs) and code validation (e.g., comparison with analytical solutions for simplified test cases, if possible), either in the main text or supplementary materials.

Minor Points:

1. Boundary condition justification: Is it possible to provide a more explicit physiological justification for the choice of boundary conditions, particularly the no-flux Neumann conditions for concentrations at the domain edges?

2. The bottom panels of Figs. 4, 7, 9 – 13 show two curves of the temporal evolution of the solutions in two spatial points. The location of these points is described in words in sec. 2.6 but it is a bit complex. It would be better for the reader to mark these two points in a figure, for example in Fig. 3 or in the first top-left panel of Fig. 4.

3. The computing time for Figures 9 -13 was 6 hours total or for each figure?

4. Regarding iterative solvers for the EMI model, in addition to [19] it would be more comprehensive to also cite the recent work by Huynh, Ngoc Mai Monica, et al. "Convergence analysis of BDDC preconditioners for composite DG discretizations of the cardiac cell-by-cell model," SIAM Journal on Scientific Computing 45.6 (2023): A2836-A2857.

**Have the authors made all data and (if applicable) computational code underlying the findings in their manuscript fully available?**

Reviewer #1: Yes

Reviewer #2: Yes

Reviewer #3: None

PLOS authors have the option to publish the peer review history of their article (what does this mean?). If published, this will include your full peer review and any attached files.

Reviewer #1: **Yes: **Blas Echebarria

Reviewer #2: No

Reviewer #3: No

**Figure resubmission:**
---

## [Editor Report · Decision Letter 1]

19 May 2025

Dear Dr. Jæger,

We are pleased to inform you that your manuscript 'Electrodiffusion dynamics in the cardiomyocyte dyad at nano-scale resolution using the Poisson-Nernst-Planck (PNP) equations' has been provisionally accepted for publication in PLOS Computational Biology.

Best regards,

Christopher E Miles

Academic Editor

PLOS Computational Biology

Feilim Mac Gabhann

Editor-in-Chief

PLOS Computational Biology

---

## [Editor Report · Acceptance letter]

PCOMPBIOL-D-25-00509R1

Electrodiffusion dynamics in the cardiomyocyte dyad at nano-scale resolution using the Poisson-Nernst-Planck (PNP) equations

Dear Dr Jæger,

I am pleased to inform you that your manuscript has been formally accepted for publication in PLOS Computational Biology. Your manuscript is now with our production department and you will be notified of the publication date in due course.

With kind regards,

Judit Kozma
